# Learning Generalizable Skills from Offline Multi-Task Data for Multi-Agent Cooperation

**Sicong Liu, Yang Shu,**\* **Chenjuan Guo, Bin Yang**
East China Normal University
`Sicongliu1014@gmail.com, {yshu,cjguo,byang}@dase.ecnu.edu.cn`

## Abstract

Learning cooperative multi-agent policy from offline multi-task data that can generalize to unseen tasks with varying numbers of agents and targets is an attractive problem in many scenarios. Although aggregating general behavior patterns among multiple tasks as skills to improve policy transfer is a promising approach, two primary challenges hinder the further advancement of skill learning in offline multi-task MARL. Firstly, extracting general cooperative behaviors from various action sequences as common skills lacks bringing cooperative temporal knowledge into them. Secondly, existing works only involve common skills and can not adaptively choose independent knowledge as task-specific skills in each task for fine-grained action execution. To tackle these challenges, we propose **Hi**erarchical and **S**eparate **S**kill **D**iscovery (HiSSD), a novel approach for generalizable offline multi-task MARL through skill learning. HiSSD leverages a hierarchical framework that jointly learns common and task-specific skills. The common skills learn cooperative temporal knowledge and enable in-sample exploration for offline multi-task MARL. The task-specific skills represent the priors of each task and achieve a task-guided fine-grained action execution. To verify the advancement of our method, we conduct experiments on multi-agent MuJoCo and SMAC benchmarks. After training the policy using HiSSD on offline multi-task data, the empirical results show that HiSSD assigns effective cooperative behaviors and obtains superior performance in unseen tasks. Source code is available at `https://github.com/mooricAnna/HiSSD`.

## 1 Introduction

Cooperative multi-agent reinforcement learning (MARL) has drawn great attention to many attractive problems such as games, intelligent warehouses, automated driving, and social science (Vinyals et al., 2019; Yun et al., 2022; Gronauer & Diepold, 2022; Tian et al., 2025; Shentu et al., 2025; Wu et al., 2025). When it comes to large-scale tasks, the MARL method yields superior performance compared to the traditional control techniques. In most real-world applications, however, building high-fidelity simulators or deploying online interaction can be costly or even infeasible. Meanwhile, multi-agent systems are expected to perform flexibility among tasks with varying numbers of agents and targets. To address these issues, training multi-agent policies that can transfer across tasks with various numbers of agents under limited experience has become an attractive direction to tackle real-world multi-agent applications (Wu et al., 2019; Kumar et al.).

Although training the multi-agent policy on a single task and fine-tuning on the target task is a simple way for policy transfer, it has the following drawbacks (Wen et al., 2022; Hu et al., 2021; Yang et al., 2022; Long et al., 2020): (i) the fine-tuning stage still requires costly interaction. (ii) it lacks the capacity to handle tasks with various numbers of agents and targets. To overcome these issues, existing works leverage Transformer (Vaswani et al., 2017) to enable a flexible population-invariant framework (Long et al., 2020; Wang et al., 2020). They also discover general cooperative behavior patterns as common skills from offline multi-task data to improve multi-agent policy transfer. ODIS

---

\*Corresponding author

(Zhang et al., 2023) conducts a two-stage offline multi-task MARL to discover generalizable multi-agent common skills. They pre-train the common skills from a global view and then optimize the policy by discovering the value-maximized skills on the multi-task data. HyGen (Zhang et al., 2024) integrates online and offline learning to ensure both multi-task generalization and training-efficiency. These methods obtained convincing improvement by learning generalizable common skills and reduced interaction costs during policy transfer.

However, existing works only learn general and reusable cooperation behaviors by aggregating cooperative actions from multi-task data. Equipping offline multi-task MARL with skill learning to improve policy transfer remains an issue. Firstly, extracting general cooperative behaviors from various cooperative actions as common skills lack bringing cooperative temporal knowledge into them. Existing works have demonstrated the significance of learning temporal knowledge in multi-agent cooperation (Xu et al., 2022b; Song et al., 2023). Secondly, existing works in literature mainly focus on discovering task-irrelevant common knowledge. Yet, few of them consider learning task-specific knowledge which is also beneficial for policy transfer in offline multi-task MARL (Yang et al., 2024; Xu et al., 2022a; Bose et al., 2024; Ishfaq et al., 2024).

In light of these issues, we propose **Hi**erarchical and **S**eparate **S**kill **D**iscovery (HiSSD), a novel approach for generalizable offline multi-task MARL through skill learning. HiSSD distinguishes knowledge derived from multi-task data into common skills and task-specific skills, utilizing a hierarchical framework that concurrently learns both categories. Concretely, the common skill represents the general cooperation patterns that involve cooperative temporal knowledge and enable in-sample dynamics exploration for offline multi-task MARL. The task-specific skill represents the unique knowledge of action execution in different tasks and achieves a task-guided fine-grained action execution. Therefore, HiSSD effectively bridges offline multi-agent policy improvement and adaptive multi-task action execution with common and task-specific skills learning.

Overall, our contributions can be summarized as following points: (i) We present HiSSD, an offline multi-task MARL method that leverages the hierarchical framework and jointly learns common and task-specific skills. (ii) HiSSD is proposed to learn common skills representing cooperative behaviors among multiple tasks for offline multi-agent policy exploration and action guidance. (iii) Meanwhile, HiSSD adaptively abstracts task-specific skills for each task to achieve a task-guided fine-grained imitation. (iv) We conduct experiments on the SMAC and multi-agent MuJoCo benchmarks. After training policy using HiSSD on offline multi-task data, the empirical results show that HiSSD assigns effective cooperative behaviors and obtains superior performance in unseen tasks.

## 2 PRELIMINARIES

### 2.1 COOPERATIVE MULTI-AGENT REINFORCEMENT LEARNING

Cooperative multi-agent reinforcement learning is formulated as a decentralized partially observable Markov decision process (Dec-POMDP) (Oliehoek et al., 2016), as the problem is defined by a tuple $\mathcal{G} = \langle K, \mathcal{S}, \Omega, \mathcal{A}, \mathcal{P}, O, \mathcal{R}, \gamma \rangle$. Here, $K = \{1, ..., k\}$ is the set of agents. The global observation $s \in \mathcal{S}$ is unobservable to each agent in the centralized training and decentralized execution (CTDE) pipeline. Cooperative agent $k$ uses local observation $o_k \in \Omega$ drawn from the observation function $O(s, k)$ to sample actions $a_k$ from the actions space $\mathcal{A}$. During interaction, the joint action $\mathbf{u} = \{a_1, a_2, ..., a_k\}$ leads to a next state $s' \sim \mathcal{P}(s'|s, \mathbf{u})$ and a global reward $r$. The training goal is to learn a cooperative multi-agent system to maximize the cumulative reward $\mathcal{R}$. We use $\boldsymbol{\tau} = \{\tau_1, \tau_2, ..., \tau_k\}$ to denote the trajectory of each agent, and the policy evaluation used to estimate the performance of the joint policy $\boldsymbol{\pi}(\mathbf{u}|\boldsymbol{\tau})$ is normally defined by rewards in infinite-horizon tasks,

$$\mathbb{E}\left[\mathcal{R}\right] = \mathbb{E}\left[\sum_{t=0}^{\infty} \gamma^t r_t(s_t, \mathbf{u}_t, s_{t+1}|\boldsymbol{\pi})\right], \tag{1}$$

where $\gamma \in [0, 1)$ is the discount factor, $r_t$ denotes the reward at time $t$.

### 2.2 LEARNING GENERALIZABLE POLICY FROM OFFLINE MULTI-TASK DATA

While cooperative multi-agent reinforcement learning has achieved advancement in many scenarios, it is still hard to transfer the policy to unseen tasks without additional interaction (Hu et al., 2021;

Iqbal et al., 2021). An effective solution is leveraging multi-task learning, extracting generalizable knowledge across tasks to improve policy transfer. In multi-task learning, tasks within the same task set possess identical types of units while exhibiting varying distributions. For instance, the quantity of agents and targets may differ across these tasks. Denote $\mathcal{M}$ as overall tasks, the whole data $\mathcal{D}^{\mathcal{M}}$ are divided into source task data $\mathcal{D}^{\mathrm{Source}}$ (or $\mathcal{D}^{\mathcal{T}}$) and target task data $\mathcal{D}^{\mathrm{Target}}$, where the target tasks are unseen during training. The goal is to train the multi-agent policy on the source task data that can be transferred to unseen tasks in the same task set without additional interaction.

## 3  METHOD

In this section, we introduce the Hierarchical and Separable Skill Discovery (HiSSD) framework, a novel approach for addressing offline multi-task multi-agent reinforcement learning problems. Our main solution is leveraging the hierarchical skill learning framework and jointly learning common and task-specific skills among multiple cooperation tasks. We begin by illustrating the overall framework of our offline multi-task MARL. We then detail the high-level planner with common skills and the low-level controller with task-specific skills. Finally, we describe the overall objective and training pipeline.

### 3.1  OFFLINE MULTI-TASK MARL WITH COOPERATIVE SKILL LEARNING

Skill is a series of latent variables representing general and reusable knowledge among tasks to guide action execution (Zhang et al., 2023; 2024). Besides the solution proposed by existing works, we give two insights into multi-task skill learning for further advancement in policy transfer. Firstly, integrating cooperative temporal knowledge into common skills helps decision-making. It gives both dynamics transition information and a global perspective into multi-agent policy. Secondly, learning task-specific skills to guide action execution is beneficial to transfer policy adaptively. It brings each task's unique knowledge into the controller and adjusts the output action distribution. In this way, we propose an offline multi-task MARL method that jointly learns common and task-specific skills to improve policy transfer. Figure 1 provides a brief illustration of the proposed framework.

Specifically, our method can be divided into two parts: (a) The high-level planner and (b) The low-level controller. The high-level planner contains a common skills encoder $\pi_{\theta_h}$, a forward predictor $f_\phi$, and a value net $V_\xi^{\mathrm{tot}}$. We feed local observation $o_t^{1:K}$ into $\pi_{\theta_h}$ to extract common skills $c_t^{1:K}$. $f_\phi$ receives common skills to output the predicted next global state $s'_{t+1}$ and local information $l_{t+1}^{1:K}$. The local information can be seen as the local observation's embedding. The value net receives $o_t^{1:K}$ and $l_t^{1:K}$ to approximate the accumulated reward $\sum \gamma r_t$. The low-level controller includes a task-specific skills encoder $g_\omega$ and an action decoder $\pi_{\theta_l}$. The task-specific skills encoder infers task-specific skills $z_t^{1:K}$ using local observation $o_t^{1:K}$. The action decoder utilizes local observation, common skills, and task-specific skills to generate the real actions $\{a_t'^k \sim \pi_{\theta_l}(\cdot|o_t^k, c_t^k, z_t^k)\}_{k=0}^K$.

Integrating cooperative temporal knowledge into common skills indicates that these skills maintain perceptibility with respect to global dynamic transitions. The transition falls into two parts, the global state transition and the value estimation. Therefore, the common skill is trained to minimize the prediction error to the real next global state $s_{t+1}$ and maximize the accumulated reward. This training objective facilitates offline exploration while effectively integrating cooperation temporal knowledge into generalizable skills through a local-to-global transition prediction mechanism. As for achieving an adaptive policy transfer, the major request is to distinguish each task's specific knowledge. The task-specific skills are trained through distribution matching with each task priors, enabling adaptive action execution guidance across diverse tasks. Moreover, our method does not require global information during execution, which differs from previous work (Liu et al., 2021). We will describe the details of the proposed skill learning in sections 3.2 and 3.3.

### 3.2  LEARNING HIGH-LEVEL PLANNER WITH COMMON SKILLS

In this subsection, we introduce a high-level planning framework designed for extracting transferable skill representations from offline multi-task MARL datasets. We start with the probabilistic inference in MARL and construct a training objective to integrate cooperative temporal knowledge into common skills.

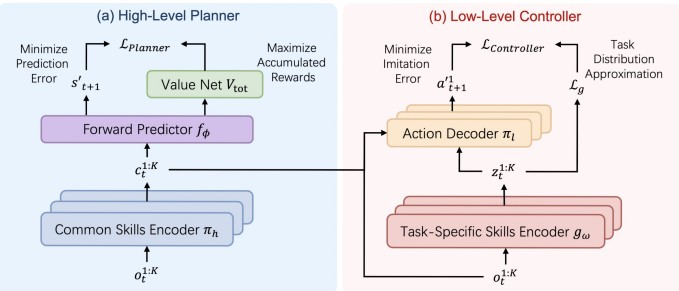

Figure 1: Overall framework of Hi-SSD. HiSSD utilizes a hierarchical framework that jointly learns common and task-specific skills from offline multi-task data to improve multi-agent policy transfer. (a) The high-level planner with common skills. HiSSD integrates cooperative temporal knowledge into common skills and enables an offline exploration. (b) The low-level controller with task-specific skills. The task-specific knowledge can guide the action execution adaptively among tasks. HiSSD uses an implicit Q-learning objective to train the value network.

Inspired by Levine (2018), we define the problem of discovering the optimal high-level planner with its common skill encoder $\pi_h^\star$ as match the trajectory $p(\tau)$ given by Eq. 2. Here, $p(\tau)$ indicates to maximize $K$ agents' accumulated reward $\sum \gamma r_t$ in each transition $p(s_{t+1}|o_t^{1:K}, c_t^{\star 1:K})$,

$$p(\tau) = \left[p(s_1)\prod_{t=1}^{T} p(s_{t+1}|o_t^{1:K}, c_t^{1:K})\right] \exp\left(\sum_{t=1}^{T} r(s_t, c_t^{1:K})\right). \quad (2)$$

where $s_t$ denotes the global state at time step $t$, $o_t^{1:K}$ indicate all agents' local observation, and $c_t^{1:K}$ represent common skills generated by the encoder $c_t^{1:K} \sim \pi_h^\star$. Matching $p(\tau)$ means that the common skill encoder $\pi_{\theta_h}$ in our learned planner needs to generate common skills $c_t^{1:K}$ and roll out trajectories $\hat{p}(\tau)$ that minimize the KL-divergence $D_{\mathrm{KL}}(\hat{p}(\tau)\|p(\tau))$. Meanwhile, learning from offline data requires the planner to be constrained and conservative. The common skill generated by the planner must lead to the next state close to the offline dataset's distribution. Therefore we conduct the learned planner as a one-step predictor and formulate $\hat{p}(\tau)$ by,

$$\hat{p}(\tau) = \left[p(s_1)\prod_{t=1}^{T} p(s_{t+1}|o_t^{1:K}, c_t^{1:K})\right] \prod_{t=1}^{T} q(s'_{t+1}|c_t^{1:K}), \quad (3)$$

where $c_t^{1:K} \sim \pi_{\theta_h}(o_t^{1:K})$ is the common skill inferred by the learned planner. $s_{t+1}$ represent the ground-truth global next state and $q(s'_{t+1}|c_t^{1:K})$ indicates that we use a forward predictor $q$ to predict the next global state using all agents' common skills. In this way, we could derive the KL-divergence and formulate our objective as below,

$$\mathcal{L}(\theta_h, \phi) = -D_{\mathrm{KL}}(\hat{p}(\tau)\|p(\tau)) = \mathbb{E}_{\substack{(o_t^{1:K}, s_{t+1})\sim\mathcal{D}^\mathcal{T}, \\ c_t^{1:K}\sim\pi_{\theta_h}(o_t^{1:K})}} \left[\sum_{t=1}^{T} \underbrace{r(s_t, c_t^{1:K})}_{\text{Exploration}} - \underbrace{\log q(s'_{t+1}|c_t^{1:K})}_{\text{Prediction}}\right], \quad (4)$$

where $\theta_h$ denote the parameters of the common skill encoder $\pi_h$ in the planner, $\mathcal{D}^\mathcal{T}$ represents the multi-task dataset. The full derivation can be found in Appendix A.1. Eq. 4 is deployed on all source tasks and aims to extract reusable cooperation knowledge as common skills $c_t^{1:K}$. This objective divides the trajectory matching into a trade-off between exploration and prediction which fulfills the requirement of integrating cooperative temporal knowledge into common skills. The *Exploration* term guarantees a value-maximization perspective in common skills to guide the action execution. The *Prediction* term not only achieves a conservative planner in offline learning but also brings the global state information into common skills, which is different from existing works.

To approximate each term in Eq. 4, we introduce a forward predictor $f_\phi$ for one-step global state prediction and a value net $V_\xi^{\mathrm{tot}}$ for reward estimation. $f_\phi$ receives common skills $c_t^{1:K}$ to predict the next global state $s'_{t+1}$ and next local information $l_{t+1}^{1:K}$. The local information can be seen as the local observation's embedding. $V_\xi^{\mathrm{tot}}$ uses the current observation $o_t^{1:K}$ or the local information $l^{1:K}$

to estimate the accumulated reward. The goal of Eq. 4 is to maximize the estimated reward and minimize the prediction error. To train the value net from the offline dataset, we utilize the Implicit Q-learning objective (Kostrikov et al., 2021) given by,

$$\mathcal{L}_{\text{IQL}}(\xi) = \mathbb{E}_{\mathcal{D}^{\mathcal{T}}} \left[ L_2^{\epsilon} \left( r_t + \gamma \bar{V}_{\bar{\xi}}^{\text{tot}}(o_{t+1}^{1:k}) - V_{\xi}^{\text{tot}}(o_t^{1:k}) \right) \right], \text{where } L_2^{\epsilon}(n) = |\epsilon - \mathbb{1}(n < 0)| n^2, \quad (5)$$

where $\epsilon \in (0, 1)$ and the target value function $\bar{V}_{\bar{\xi}}^{\text{tot}}$ is the momentum version of $V_{\xi}^{\text{tot}}$, $\mathcal{D}^{\mathcal{T}}$ is the multi-task dataset. This objective down weights the contributions of the TD-residual smaller than $0$ while giving more weight to larger values. By substituting $f_\phi$ and $V_{\xi}^{\text{tot}}$ into Eq. 4, we could rewrite the empirical objective for learning the high-level planner with common skills as below,

$$\mathcal{L}_{\text{Planner}}(\theta_h, \phi) = \mathbb{E}_{\substack{(o_t^{1:K}, s_{t+1}) \sim \mathcal{D}^{\mathcal{T}}, \\ c_t^{1:K} \sim \pi_{\theta_h}(o_t^{1:K})}} \left[ V_{\xi}^{\text{tot}}(l_{t+1}^{1:K}) - \alpha \log f_\phi(s_{t+1}'|c_t^{1:K}) \right], \quad (6)$$

where $l_{t+1}^{1:K} \sim f_\phi(\cdot|c_t^{1:K})$ is the predicted local information. The weight $\alpha$ serves as the trade-off between guiding to space with high-reward and space that the execution policy ought to have a correct imitation. Inspired by Xu et al. (2022a), we introduce an alternative objective that implicitly involves the behavior constraint by using the TD-residual $[r + \gamma \bar{V}_{t+1}^{\text{tot}} - V_t^{\text{tot}}]$ as the imitation weight,

$$\mathcal{L}_{\text{Planner}}(\theta_h, \phi) = \mathbb{E}_{\substack{(o_t^{1:K}, s_{t+1}) \sim \mathcal{D}^{\mathcal{T}}, \\ c_t^{1:K} \sim \pi_{\theta_h}(o_t^{1:K})}} \left[ \exp \left( \frac{r + \gamma \bar{V}_{\xi}^{\text{tot}}(l_{t+1}^{1:K}) - V_{\xi}^{\text{tot}}(o_t^{1:K})}{\alpha} \right) \log f_\phi(s_{t+1}'|c_t^{1:K}) \right], \quad (7)$$

Following this objective, the common skill acquires a global cooperative perspective and is more likely to represent behavior patterns with high rewards from offline multi-task data.

## 3.3 LEARNING LOW-LEVEL CONTROLLER WITH TASK-SPECIFIC SKILLS

After constructing the objective of common skills, we now turn to the low-level controller with task-specific skills $z_i^{1:K}$, where $i \in \mathcal{T}$ denotes the current task. Here we omit the subscript of timestep $t$ to simplify the notation. The main function of our proposed controller is to generate real actions following the skills' guidance. Meanwhile, the job of the task-specific skill is to recognize the current task and guide the policy adaptively. To be more specific, we denote the low-level controller's components task-specific skills encoder as $g_\omega(z_i^{1:K}|o_i^{1:K})$ and the action decoder $\pi_{\theta_l}(a_i^{1:K}|z_i^{1:K}, o_i^{1:K}, c_i^{1:K})$. We adopt an objective based on $\beta$-VAE (Higgins et al., 2017) to learn action execution and utilize Self-Supervised Learning to learn task-specific representations for regularization. This learning objective is given by,

$$\mathcal{L}_{\text{Controller}}(\theta_l, \omega) = -\mathbb{E}_{\substack{\mathcal{D}^i \in \mathcal{T} \\ z_i \sim g_\omega(o_i)}} \left[ \log \pi_{\theta_l}(a_i^{1:K}|o_i^{1:K}, c_i^{1:K}, z_i^{1:K}) - \beta D_{\text{KL}}(z_i^{1:K} \| p(\mathcal{D}^i)) \right], \quad (8)$$

where the $\mathcal{T}$ denotes the task number, $i$ denotes the current task, $\theta_l$ and $\omega$ represent the parameters of action decoder $\pi_{\theta_l}$ and task-specific skills encoder $g_\omega$ respectively, and $\beta$ is the regularization coefficient. The left term requires the action decoder $\pi_{\theta_l}$ to maximize the likelihood of the real action $a_i^{1:K}$ from offline data, the right term minimizes the KL-divergence between the task-specific skills $z_i^{1:K}$ and the task priors $p(\mathcal{D}^i)$ as a regularization. We show the full derivation in Appendix A.2. Notably, the task-specific skill $z_i^{1:K}$ is regularized to approach the current task priors $p(\mathcal{D}^i)$. We show that the problem can be formulated as a contrastive learning objective, with its theoretical lower bound provided in Theorem 3.1. Training the skill encoder for task discrimination enables the integration of task-specific knowledge into the skill embeddings $z_i^{1:K}$.

**Theorem 3.1.** Denote a set of $N$ training tasks with their offline data $\mathcal{D}^{\mathcal{T}}$, $\mathcal{D}^i$ is the data of the sampled task. Let random variables $x$ be some observations sampled from $\mathcal{D}^i$, skill $z \sim g_\omega(z|x)$, $h(x, z) = \frac{g_\omega(z|x)}{p(z)}$, $p(\mathcal{D}^i)$ is the prior distribution of current task $i$, then we have

$$-\mathbb{E}_{\mathcal{D}^i, z, x} \left[ \log \frac{h(z, x)}{\sum_{\mathcal{D}^j \in \mathcal{D}^{\mathcal{T}}} h(z, x^j)} \right] \geq -\mathbb{E}_{x \sim \mathcal{D}^i} \left[ D_{\text{KL}}(g(\cdot|x) \| p(\mathcal{D}^i)) \right] \quad (9)$$

where $x^j$ are observations sampled from task $\mathcal{D}^j$ and $i \in \{1, 2, \cdots, N\}$.

We leave the proof of Theorem 3.1 to Appendix A.3. In practice, we utilize the exponential cosine similarity $\exp(z \cdot x)$ to approximate $h(z, x)$ and randomly sample two agents' observations $\{o_i^m, o_i^n\}$ in the same task as the positive pairs $(q, k_+)$ and regard trajectories from other tasks as the negative samples $k_-$. We introduce $g_{\bar{\omega}}$ the momentum version of $g_{\omega}$ to optimize the contrastive loss,

$$\mathcal{L}_g(\omega) = -\sum_{\mathcal{D}^i}^{\mathcal{D}^{\mathcal{T}}} \mathbb{E}_{\substack{\{q,k_+\} \sim \mathcal{D}^i, \\ k_- \sim \mathcal{D}^{\mathcal{T} \setminus i}}} \left[ \log \frac{\exp(g_{\omega}(q) \cdot g_{\bar{\omega}}(k_+)/\sigma)}{\exp(g_{\omega}(q) \cdot g_{\bar{\omega}}(k_+)/\sigma) + \sum_{\mathcal{T} \setminus i} \exp(g_{\omega}(q) \cdot g_{\bar{\omega}}(k_-)/\sigma)} \right], \quad (10)$$

where $\mathcal{D}^{\mathcal{T}}$ is overall source tasks, $\mathcal{D}^i$ and $\mathcal{D}^{\mathcal{T} \setminus i}$ denote the current task and other tasks in $\mathcal{D}^{\mathcal{T}}$, respectively. $\sigma$ denotes the temperature and $g_{\bar{\omega}}$ is updated by the exponential moving average $\bar{\omega}' \leftarrow \eta\omega + (1 - \eta)\bar{\omega}$. This training pipeline empowers the policy to embed task-specific knowledge and achieve adaptive action execution.

In summary, we integrate Eqs. 9 and 10 into Eq. 8 to obtain the empirical objective of training the low-level controller. We propose to learn the prior of the current task using Eq. 10 and replace the $D_{\mathrm{KL}}$ term in Eq. 8 with it. The objective is given by,

$$\mathcal{L}_{\mathrm{Controller}}(\theta_l, \omega) = -\sum_{\mathcal{D}^i \in \mathcal{D}^{\mathcal{T}}} \mathbb{E}_{(o_i,a_i) \sim \mathcal{D}^i} \left[ \log \pi_{\theta_l}(a_i^{1:K}|o_i^{1:K}, c_i^{1:K}, z_i^{1:K}) \right] - \beta\mathcal{L}_g(\omega). \quad (11)$$

### 3.4 TRAINING AND EVALUATION

HiSSD is fully trained offline and can be trained end-to-end. During training, each task's data $\mathcal{D}^i$ in the source dataset $\mathcal{D}^{\mathcal{T}}$ will be chosen for training. In every training step, we sequentially optimize Eqs. 5, 7, and 11 to train the value net, the planner, and the controller, respectively. At the test time, we only use local information to perform decentralized execution. The planner $\pi_{\theta_h}$ infers the common skill $c_t^{1:K}$ at each time step. The controller $\pi_{\theta_l}$ first generates task-specific skill $z_t^{1:K}$ and then feed $\{(o_t^k, c_t^k, z_t^k)\}_{k=0}^K$ into action decoder to generate the real action. Due to space limitations, we leave the pseudocode in Algorithm 1 in Appendix B.

## 4 EXPERIMENTS

### 4.1 BENCHMARKS AND DATASETS

**SMAC** The StarCraft Multi-Agent Challenge (SMAC) (Samvelyan et al., 2019) is a popular MARL benchmark and can evaluate multi-task learning or policy transfer methods. We follow the experimental settings by Zhang et al. (2023) and use the offline dataset they collected. Similar to the D4RL benchmark (Fu et al., 2020), there are four dataset qualities labeled as *Expert*, *Medium*, *Medium-Expert*, and *Medium-Replay*. We construct task sets Marine-Easy and Marine-Hard. In each task set, units in different tasks have the same type and various numbers, all algorithms are trained on offline data from multiple source tasks and evaluated on a wide range of unseen tasks without additional data. Details are referred to the Appendix C.

**MAMuJoCo** Multi-Agent MuJoCo (MAMuJoCo) is a benchmark for continuous multi-agent robotic control, based on the MuJoCo environment. To fulfill the requirement of offline multi-task learning, we follow Wang et al. (2023a) and collect a multi-task dataset in *HalfCheetah-v2* using HAPPO (Kuba et al., 2022) algorithm. We partition the robotic into six agents and construct the individual task by disabling each agent. Each task's name corresponds to the joint controlled by the disabled agent. Algorithms are trained on multiple source tasks and evaluated on unseen tasks without additional data. Details of the dataset are presented in Appendix C.

### 4.2 BASELINES

**SMAC** To evaluate the capacity of policy transfer using HiSSD, we introduce several comparable baselines from prior works: (i) ODIS (Zhang et al., 2023), an effective offline multi-task MARL method for cooperative skill discovery. (ii) UPDeT-m, an offline variant of UPDeT (Hu et al., 2021) by adopting the transformer-based Q mixing network. (iii) Transformer-based behavior cloning (**BC-t**) method and its variants with return-to-go information (**BC-r**). We average Hi-SSD's performance over 5 random seeds and report the best score for each task.

Table 1: Average test win rates of the best policies over five random seeds in the task set *Marine-Hard* with different data qualities. For simplicity, the asymmetric task names are abbreviated. For example, the task name "5m6m" denotes the SMAC map "5m_vs_6m". Results of BC-best stands for the best test win rates between BC-t and BC-r.

| Tasks | Expert | | | | Medium | | | |
|---|---|---|---|---|---|---|---|---|
| | BC-best | UPDeT-m | ODIS | HiSSD (Ours) | BC-best | UPDeT-m | ODIS | HiSSD (Ours) |
| Source Tasks | | | | | | | | |
| 3m | 97.7± 2.6 | 82.8±16.0 | 98.4± 2.7 | **99.5± 0.3** | 65.4±14.7 | 51.2± 3.4 | **85.9±10.5** | 62.7± 5.7 |
| 5m6m | 50.4± 2.3 | 17.2±28.0 | 53.9± 5.1 | **66.1± 7.0** | 21.9± 3.4 | 6.3± 4.9 | 22.7± 7.1 | **26.4± 3.8** |
| 9m10m | 95.3± 1.6 | 3.1± 5.4 | 80.4± 8.7 | **95.5± 2.7** | 63.8±10.9 | 28.5±10.2 | **78.1± 3.8** | 73.9± 2.3 |
| Unseen Tasks | | | | | | | | |
| 4m | 92.1± 3.5 | 33.0±27.1 | 95.3± 3.5 | **99.2± 1.2** | 48.8±21.1 | 14.1± 5.2 | 61.7±17.7 | **77.3±10.2** |
| 5m | 87.1±10.5 | 33.6±40.2 | 89.1±10.0 | **99.2± 1.2** | 76.6±14.1 | 67.2±21.3 | 85.9±11.8 | **88.4± 8.4** |
| 10m | 90.5± 3.8 | 54.7±44.4 | 93.8± 2.2 | **98.4± 0.8** | 56.2±20.6 | 32.9±11.3 | 61.3±11.3 | **98.0± 0.3** |
| 12m | 70.8±15.2 | 17.2±28.0 | 58.6±11.8 | **75.5±19.7** | 24.0±10.5 | 3.2± 3.8 | 35.9± 8.1 | **86.4± 6.0** |
| 7m8m | 18.8± 3.1 | 0.0± 0.0 | 25.0±15.1 | **35.3± 9.8** | 1.6± 1.6 | 0.0± 0.0 | **28.1±22.0** | 14.2±10.1 |
| 8m9m | 15.8± 3.3 | 0.0± 0.0 | 19.6± 6.0 | **47.0± 6.2** | 3.1± 3.8 | 2.3± 2.6 | 4.7± 2.7 | **15.3± 2.8** |
| 10m11m | 45.3±11.1 | 0.0± 0.0 | 42.4± 7.2 | **86.3±14.6** | 19.7± 8.9 | 4.0± 3.4 | 29.7±15.4 | **43.6± 4.6** |
| 10m12m | 1.0± 1.5 | 0.0± 0.0 | 1.6± 1.6 | **14.5± 9.1** | 0.0± 0.0 | 0.0± 0.0 | **1.6± 1.6** | 0.6± 0.5 |
| 13m15m | 0.0± 0.0 | 0.0± 0.0 | **2.3± 2.6** | 1.3± 2.5 | 0.6± 1.3 | 0.0± 0.0 | **1.6± 1.6** | 1.4± 2.4 |
| | Medium-Expert | | | | Medium-Replay | | | |
| Source Tasks | | | | | | | | |
| 3m | 67.7±23.7 | 85.2±17.9 | 73.6±22.0 | **86.6± 3.7** | 81.1± 8.8 | 41.4±20.1 | **83.6±14.0** | 78.8± 5.3 |
| 5m6m | 31.3± 6.3 | 1.6± 1.6 | 9.4± 2.2 | **41.9± 9.7** | 25.0± 3.1 | 0.8± 1.4 | 16.6± 4.7 | **25.3±10.3** |
| 9m10m | 26.0±13.9 | 24.3±18.7 | 31.3±14.5 | **83.6± 6.9** | 33.4±13.1 | 0.8± 1.4 | 34.4± 8.0 | **45.8± 3.9** |
| Unseen Tasks | | | | | | | | |
| 4m | 81.3±18.9 | 43.9±39.0 | 82.8±13.5 | **91.1± 6.1** | 61.5± 9.0 | 35.9±12.6 | 55.6±14.5 | **77.3± 1.9** |
| 5m | 74.0± 2.9 | 33.6±40.2 | 82.8±17.7 | **98.3± 1.8** | 75.0±24.2 | 61.7±20.3 | **96.1± 4.1** | 88.1±13.4 |
| 10m | 78.1± 6.7 | 32.8±38.1 | 82.8±16.8 | **96.4± 2.1** | 82.4± 8.2 | 11.0± 7.8 | 84.4±15.1 | **94.7± 2.6** |
| 12m | 64.8±24.3 | 9.4± 8.6 | 81.3±20.6 | **88.4±11.8** | 83.4± 4.5 | 2.3± 2.6 | 84.4± 6.6 | **90.3± 3.6** |
| 7m8m | 13.3± 4.5 | 2.3± 4.1 | 15.6± 4.4 | **30.5±10.4** | 7.3± 6.4 | 1.6± 2.7 | 9.4± 2.2 | **21.7± 4.7** |
| 8m9m | 10.2± 4.6 | 9.5± 8.6 | 10.9± 4.7 | **35.2±18.3** | 11.5± 3.9 | 0.8± 1.4 | 11.7± 8.7 | **14.5± 4.0** |
| 10m11m | 26.6± 4.7 | 11.8± 8.1 | 33.6± 8.9 | **54.7± 6.8** | 46.8± 6.6 | 0.8± 1.4 | 35.9± 5.2 | 42.5± 4.4 |
| 10m12m | 0.0± 0.0 | 0.0± 0.0 | 1.6± 1.6 | **2.5± 1.0** | 1.6± 2.7 | 0.0± 0.0 | **2.3± 1.4** | 0.5± 0.3 |
| 13m15m | 0.8± 1.4 | 0.0± 0.0 | 2.3± 2.6 | **5.2± 3.7** | 1.6± 1.6 | 0.0± 0.0 | 2.4± 1.4 | **3.6± 2.1** |

**MAMuJoCo**    For the continuous robotic control task, we compare HiSSD with four recent offline MARL algorithms: (i) Behavior cloning (**BC**) method, the multi-agent version of (ii) IQL Kostrikov et al. (2021), (iii) TD3-BC (Fujimoto & Gu, 2021), and (iv) ODIS (Zhang et al., 2023) reproduced by ourselves. All algorithms are evaluated over 32 independent runs, with 4 different random seeds employed during training to ensure reproducibility. Notably, all algorithms use the same architecture, and the details for the implementations and hyperparameters of algorithms in MAMuJoCo are shown in Appendix D.

## 4.3    MAIN RESULTS

**SMAC**    We evaluate Hi-SSD and baselines on SMAC and present the average test win rates in the Marine-Hard task set in Table 1. BC-best represents the highest test win rates between BC-t and BC-r. Below are the key results: (i) HiSSD achieves top performance on over half of the tasks. This indicates that using skills to guide action execution benefits policy transfer. (ii) Compared to ODIS, which only discovers common skills from offline multi-task data, our method outperforms it in medium and near-optimal data qualities, showing advancement in learning task-specific skills. Due to space limitation, we leave results on other task sets in appendix E.

**MAMuJoCo**    Table 2 shows the mean and standard deviation of average returns for the offline multi-task learning on our HalfCheetah-v2 task set. The results show that HiSSD outperforms all baselines and achieves state-of-the-art performance in most tasks. Compared to baselines with only behavior cloning (BC), HiSSD outperforms in a wide range. We also find that HiSSD outperforms ODIS and TD3-BC when transferring to unseen tasks. It indicates that learning cooperative temporal knowledge and task-specific skills is beneficial for learning generalizable multi-agent policy from offline multi-task data.

Table 2: Average scores on HalfCheetah-v2 multi-task datasets in MAMuJoCo.

| Tasks | BC | IQL | ITD3-BC | ODIS | HiSSD(ours) |
|---|---|---|---|---|---|
| | | | Source Tasks | | |
| complete | 3188.16±566.68 | 4384.23±198.55 | 4365.10± 72.92 | 3677.66±174.82 | **4450.57±126.36** |
| back thigh | 3324.22± 58.49 | 3675.91± 18.99 | 3685.82± 40.84 | 2381.62±198.59 | **3698.38± 13.98** |
| back foot | 3079.01±355.66 | 3989.12±211.42 | **4119.48± 61.00** | 2713.40±195.63 | 3197.83± 6.99 |
| front thigh | 1861.53±415.80 | **2744.63±329.00** | 2700.17±407.46 | 2684.99±249.70 | 1948.74± 81.24 |
| front shin | 1819.94±273.96 | 4048.43±363.79 | **4155.15±180.99** | 3944.78±219.72 | 3468.32±290.72 |
| | | | Unseen Tasks | | |
| back shin | 1964.55±268.24 | 1974.62±314.33 | 1690.40±251.77 | 3217.59±184.15 | **3472.12± 91.95** |
| front foot | 3468.40±369.40 | 3948.17±381.80 | 3683.16±419.42 | 3930.82±342.16 | **4175.29±338.96** |

Table 3: Ablation studies on HiSSD. We report average test win rates of the best policies over five random seeds in the task set *Marine-Hard* with different data qualities.

| Data Qualities | w/o Planning | w/o Predicting | Half-Negative | L2-Loss | BC-best | HiSSD |
|---|---|---|---|---|---|---|
| | | | Source Tasks | | | |
| Expert | 80.7±22.4 | 85.9±16.4 | **87.0±15.3** | 80.8±21.2 | 81.1±21.8 | 84.5±17.1 |
| Medium | 52.7±25.5 | 53.1±20.0 | 54.3±20.6 | 42.3±21.8 | 50.3±20.1 | **55.9±19.2** |
| Medium-Expert | 56.0±22.4 | 63.5±26.1 | **70.7±21.3** | 57.8±27.1 | 41.7±18.5 | 64.2±23.5 |
| Medium-Replay | 51.1±28.2 | 47.7±25.1 | 50.0±22.9 | 44.7±28.1 | 46.5±26.7 | **51.9±24.0** |
| | | | Target Tasks | | | |
| Expert | 45.8±39.0 | 55.0±37.4 | 60.9±37.1 | 53.2±38.9 | 46.8±36.8 | **61.2±37.4** |
| Medium | 38.4±37.5 | 37.6±34.2 | 46.7±38.0 | 42.4±38.6 | 25.6±26.8 | **48.6±40.1** |
| Medium-Expert | 47.2±36.9 | 51.2±34.3 | 55.8±37.6 | 51.0±38.9 | 38.8±33.0 | **57.6±37.5** |
| Medium-Replay | 45.0±37.8 | 42.8±36.2 | 48.1±37.6 | 47.9±40.3 | 41.2±33.7 | **48.7±39.0** |

## 4.4 ABLATION STUDY

In this section, we provide additional empirical analysis of HiSSD. First, we demonstrate the effectiveness of our skill learning approach that jointly encodes cooperative temporal patterns and task-specific knowledge. Next, we identify key factors in task-specific skill acquisition. Finally, we present skill visualizations for deeper insights.

**Common Skills Analysis** We conduct experiments on SMAC's Marine-Hard task set to demonstrate the effects of our proposed skills learning paradigm in HiSSD and present results in Table 3. We implement two variants of HiSSD to show the impact of three factors on HiSSD's skills learning, respectively: (i) *w/o Planning*. HiSSD only learns task-specific skills among tasks. (ii) *w/o Prediction*. HiSSD trains the planner without the next global state prediction. The results indicate that learning common skills improves policy transfer, and learning to predict the next global state acquires further advancement.

**Task-Specific Skills Analysis** To investigate what is the essential factor of learning task-specific skills, we conduct experiments on SMAC's Marine-Hard task set with three variants of HiSSD: (i) *Half-Negative*. We reduce to half of the negative samples during contrastive learning. (ii) *L2-Loss*. The objective is replaced with one similar to Grill et al. (2020) which does not require negative samples. According to the results in 3, learning to distinguish the difference between tasks plays a key role in learning task-specific skills. Meanwhile, increasing the number of negative samples further improves the capacity of multi-agent policy transfer.

## 4.5 VISUALIZATION OF LEARNED SKILLS

To investigate the effectiveness of our proposed method more clearly, we evaluate HiSSD on multiple tasks in SMAC and visualize the learned skills using t-SNE (Hinton & Roweis, 2002).

**Common Skills** In Figure 2, we visualize the chosen common skills among multiple tasks. Neighboring points in the same distribution represent similar common skills. To show the skill flow, we partition the collected trajectories into four sub-parts by timestep. The plots show that common

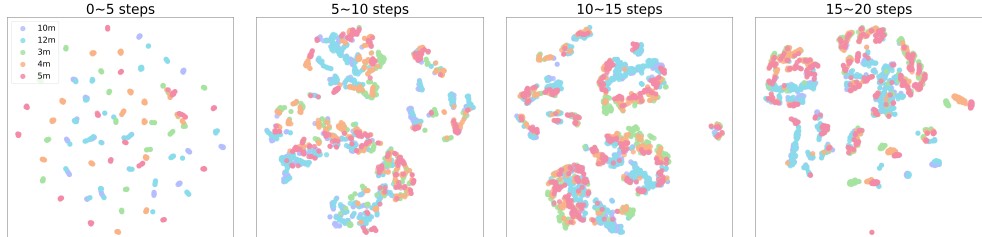

Figure 2: Visualization of learned common skills. We use HiSSD to collect trajectories on five tasks in SMAC and partition these trajectories into four time windows. Plots in each figure represent the distribution of chosen common skills. We use multiple time windows to indicate the task flow.

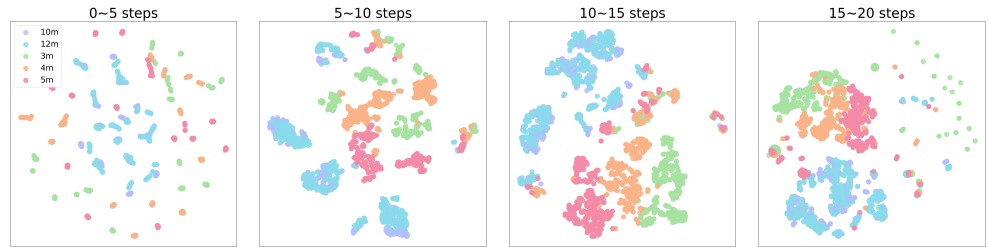

Figure 3: Visualization of learned task-specific skills. We use HiSSD to collect trajectories on five tasks in SMAC and partition these trajectories into four time windows. Plots represent the distribution of chosen task-specific skills. We use multiple time windows to indicate the task flow.

skills are mapped into multiple clusters each containing skills from different tasks. Points with the same color in each distribution represent the collaboration between agents in the corresponding task. The results indicate that HiSSD acquires the capability to learn task-irrelevant common skills.

**Task-Specific Skills** Figure 3 represented the chosen task-specific skills during evaluation. We collect trajectories on multiple tasks in SMAC using HiSSD and partition these trajectories into four sub-parts by timestep. From the plots, task-specific skills chosen in small-scale tasks (i.e., *3m*, *4m*, and *5m*) are mapped into different distributions. However, the chosen skills in large-scale tasks (i.e., *10m* and *12m*) overlap with each other. The results indicate that HiSSD efficiently generalizes to small-scale tasks similar to source domains, learning task discrepancies effectively. However, for large-scale tasks (10m, 12m), it struggles to capture significant distribution shifts, leading to overlapping. Notably, our method adaptively reduces distribution distance over time, demonstrating dynamic transition learning across tasks. As episodes progress, task representations converge due to agent attrition, where multi-agent tasks naturally decompose into simpler sub-tasks during episode execution. This demonstrates HiSSD's ability to: (i) effectively encode task-specific knowledge, and (ii) adaptively distinguish between tasks through learned policies.

## 5 RELATED WORK

### 5.1 OFFLINE MARL

Training policies from offline experience without interaction effectively reduce the trial and error costs when implementing RL in real-world scenarios (Levine et al., 2020; Zhang et al., 2021). Due to the distribution shift in offline learning (Fujimoto et al., 2019), training a policy from static datasets faces unexpected extrapolation errors when estimating unseen data (Wu et al., 2019; Kumar et al.). Therefore, previous works consider learning behavior-constrained policies (Kumar et al., 2020; Kostrikov et al., 2021), which can be extended to the MARL paradigm. They aim at adopting sufficient conservatism to current online MARL methods (Yang et al., 2021; Jiang & Lu, 2021; Pan et al., 2022), training policies by value function decomposition (Sunehag et al., 2018; Rashid et al., 2020; Yang et al., 2020b; Wang et al., 2021), or multi-agent policy gradient algorithms (Lowe et al., 2017; Foerster et al., 2018; Iqbal & Sha, 2019; Yu et al., 2022; Kuba et al., 2022). Another effective

way for offline learning is leveraging the powerful transformer-based model (Chen et al., 2021; Lee et al., 2022; Meng et al., 2023; Liu et al., 2023) or diffusion model (Janner et al., 2022; Pearce et al., 2023; He et al., 2023). Yet the problem of combining generative models with the policy improvement paradigm of reinforcement learning remains an issue (Zheng et al., 2022; Wang et al., 2023b; Kang et al., 2023).

## 5.2 MULTI-TASK MARL

Multi-task learning plays a key role in improving data-efficiency and generalization in MARL. It highlights knowledge reuse (Da Silva & Costa, 2016; Shen et al., 2021; Sodhani et al., 2021), which is beneficial for transferable multi-agent collaboration. This paradigm requires the policy to hold a flexible structure for deploying agents across tasks with varying input dimensions (Agarwal et al., 2020; Wang et al., 2020; Hu et al., 2021; Zhou et al., 2021). Recent works consider multiple ways to realize multi-task adaptations such as policy representations learning (Grover et al., 2018), evolutionary-based curriculum learning (Long et al., 2020), randomized entity-wise factorization (Iqbal et al., 2021), high-level cooperation strategy reusing (Liu et al., 2021), and training transformer-based population-invariant policies (Hu et al., 2021; Wen et al., 2022; Liu et al., 2023). Although these methods relieve the need for learning from scratch during transferrin, generalizing policies without simultaneous learning or fine-tuning remains challenging Zhang et al. (2023).

## 5.3 MARL WITH SKILL LEARNING

Hierarchical MARL with skill learning is a practical way to solve complex decision-making tasks. This paradigm embeds behavior patterns in a skill space, promoting the exploration of cooperative multi-agents with state empowerment from information theory (Barto & Mahadevan, 2003; Eysenbach et al., 2019; Yang et al., 2023; He et al., 2020; Liu et al., 2022). MASD (He et al., 2020) introduces an information bottleneck for cooperation patterns discovery. HSD (Yang et al., 2020a) and HSL (Liu et al., 2022) utilize hierarchical architectures to discover diverse behaviors. HMASD (Yang et al., 2023) treats skill discrimination as a sequential modeling problem. However, their framework requires global information during execution. VO-MASD (Chen et al., 2024) discovers hierarchy-like cooperation skills in a pre-training stage to speed up the online learning. ODIS (Zhang et al., 2023) combines offline multi-task learning with hierarchical MARL to learn a generalizable multi-agent policy. Although they discover skills following the value function decomposition, they only consider the common skill among tasks. HyGen (Zhang et al., 2024) follows ODIS and integrates online exploration to further improve transfer capability, especially in middle data qualities. In this article, we propose a hierarchical policy that jointly learns common and task-specific skills from offline multi-task data, further enhancing the capacity of multi-agent policy's generalization.

## 6 CONCLUSION

In this paper, we propose a new hierarchical multi-agent policy that jointly learns common and task-specific skills from offline multi-task data, further improving the capacity of policy transfer in offline multi-task MARL. We analyze the primary issues in current offline multi-task MARL methods and propose novel objectives to overcome these issues. We compare HiSSD to SOTA methods on popular MARL benchmarks and certify that it acquires promising improvement. One limitation of Hi-SSD is its training stability, and we consider it as our future work. Hopefully, our proposed skill learning pipeline can lead to a new branch for offline multi-task learning in MARL.

## 7 REPRODUCIBILITY STATEMENT

Source code is available at `https://github.com/mooricAnna/HiSSD`. The full derivation of our proposed objective and theorem is presented in Appendix A. The pseudocode is presented in Appendix B. We leave the detailed description of the used benchmarks and datasets in Appendix C and present the implementation details in Appendix D.

# 8 ACKNOWLEDGMENT

This work was supported by National Natural Science Foundation of China (62406112, 62372179).

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

# A ADDITIONAL DERIVATION

## A.1 OBJECTIVE FOR HIGH-LEVEL PLANNER

In this section, we formulate our objective using probabilistic inference, following the theoretical analysis in Levine (2018). We assume that the training goal for behavior cloning is to minimize the distance between the optimal trajectory $p(\tau)$ formulated in Eq. 2 and the rollout trajectory $\hat{p}(\tau)$. $\hat{p}(\tau)$ is required to predict the next global state $s'_{t+1}$ that is not far from the offline dataset. Therefore, we could formulate $\hat{p}(\tau)$ as below,

$$\hat{p}(\tau) = \left[ p(s_1) \prod_{t=1}^{T} p(s_{t+1}|o_t^{1:K}, c_t^{1:K}) \right] \prod_{t=1}^{T} q(s'_{t+1}|c_t^{1:K}), \tag{12}$$

where $q(\cdot)$ is a predefined predictor, $c_t^{1:K}$ is the learned common skill. A practical way to match this objective is by adopting an optimization-based approximate inference approach. In this article, we minimize the KL-divergence between the approximated trajectory and the optimal trajectory to achieve this objective,

$$D_{\mathrm{KL}}(\hat{p}(\tau)\|p(\tau)) = -\mathbb{E}_{\tau\sim\hat{p}(\tau)}\left[\log p(\tau) - \log \hat{p}(\tau)\right]. \tag{13}$$

In this way, we derive Eq. 13 and obtain the planner's objective,

$$-D_{\mathrm{KL}}(\hat{p}(\tau)\|p(\tau)) = \mathbb{E}_{\tau\sim\mathcal{D}}\left[\log p(\tau) - \log \hat{p}(\tau)\right]$$

$$= \mathbb{E}_{\tau\sim\mathcal{D}}\left[\log p(s_1) + \sum_{t=1}^{T}(\log p(s_{t+1}|o_t^{1:K}, c_t^{1:K}) + r(s_t, c_t^{1:K})) - \right.$$

$$\left. \log p(s_1) - \sum_{t=1}^{T}(\log p(s_{t+1}|o_t^{1:K}, c_t^{1:K}) + \log q(s'_{t+1}|o_t^{1:K}, c_t^{1:K})) \right]$$

$$= \mathbb{E}_{\tau\sim\mathcal{D}}\left[\sum_{t=1}^{T} r(s_t, c_t^{1:K}) - \log q(s'_{t+1}|o_t^{1:K}, c_t^{1:K}) \right]$$

$$= \sum_{t=1}^{T} \mathbb{E}_{\tau\sim\mathcal{D}}\left[r(s_t, c_t^{1:K}) - \log q(s'_{t+1}|o_t^{1:K}, c_t^{1:K})\right]. \tag{14}$$

## A.2 OBJECTIVE FOR LOW-LEVEL CONTROLLER

To formulate the objective for our fine-grained action controller, we perform the controller as a generative model and introduce the variational inference. With the help of Jensen's inequality, we formulate the ELBO as below,

$$\log p(a) = \log \int_z p(a, z)\frac{q(z|o)}{q(z|o)} = \log \mathbb{E}_{q(z|o)}\left[\frac{p(a, z)}{q(z|o)}\right]$$

$$\geq \mathbb{E}_{q(z|o)}\left[\log p(a, z)\right] - \mathbb{E}_{q(z|o)}\left[\log q(z|o)\right]$$

$$= \mathbb{E}_{q(z|o)}\left[p(a|z) \cdot p(z)\right] - \mathbb{E}_{q(z|o)}\left[\log q(z|o)\right] \to \mathrm{ELBO}, \tag{15}$$

where $q(\cdot)$ is a predefined encoder, $o$ denotes the local observation of each agent, and $z$ denotes the task embedding. Practically, we introduce a task-guided controller $\pi_{\theta_l}$ as $p(a|z)$ and a task discriminator $g_\omega$ as $q(z|o)$. The prior distribution $p(z)$ is unreachable to each task during training and we use $p(\mathcal{D}^i)$ to represent the prior distribution of skill $z$ in each task. We also embed the common skill $c_t$ and local observation $o_t$ into $\pi_{\theta_l}$. Thus, we rewrite Eq. 15 and formulate the objective of our low-level controller in a decentralized manner,

$$\mathcal{L}_{\mathrm{Controller}}(\theta_c, \omega) = -\mathcal{L}_{\mathrm{ELBO}}$$

$$= -\mathbb{E}_{\mathcal{D}^i\sim\mathcal{D}}\left[\mathbb{E}_{g_\omega(z_t^i|o_t^i)}\left[\log \pi_{\theta_l}(a_t^i|z_t^i, o_t^i, c_t^i)\right] + \mathbb{E}_{g_\omega(z_t^i|o_t^i)}\left[\log \frac{p(\mathcal{D}^i)}{g_\omega(z_t^i|o_t^i)}\right]\right]$$

$$= -\mathbb{E}_{\mathcal{D}^i\sim\mathcal{D}}\left[\mathbb{E}_{g_\omega(z_t^i|o_t^i)}\left[\log \pi_{\theta_l}(a_t^i|z_t^i, o_t^i, c_t^i)\right] - D_{\mathrm{KL}}(z_t^i\|p(\mathcal{D}^i)))\right], \tag{16}$$

where $i$ denotes the current task and $\mathcal{D}$ indecates the offline d ataset.

### A.3 BRIDGING KL-DIVERGENCE AND CONTRASTIVE LEARNING

In this section, we illustrate how to bridge the contrastive loss to approximate the KL-divergence in Eq. 8 We first introduce a lemma. Then we give a proof of Theorem 3.1.

**Lemma A.1.** Given $\mathcal{D}^i \sim \mathcal{D}^{\mathcal{T}}$ as the the current task's data distribution. Denote $x$ as the local observations sampled from the offline data $x \sim \mathcal{D}^i$, $g_\omega$ is the task-specific skill encoder, skill $z \sim g_\omega(z|x)$. Then we have,

$$\frac{p(z|\mathcal{D}^i)}{p(z)} = \int \frac{p(x|\mathcal{D}^i)p(z|x,\mathcal{D}^i)}{p(z)}\mathrm{d}x = \int \frac{p(x)g_\omega(z|x)}{p(z)}\mathrm{d}x = \mathbb{E}_x\left[\frac{g_\omega(z|x)}{p(z)}\right]. \qquad (17)$$

**Theorem 3.1.** Denote a set of $N$ training tasks with their offline data $\mathcal{D}^{\mathcal{T}}$, $\mathcal{D}^i$ is the data of the sampled task. Let random variables $x$ be some observations sampled from $\mathcal{D}^i$, skill $z \sim g_\omega(z|x)$, $h(x,z) = \frac{g_\omega(z|x)}{p(z)}$, $p(\mathcal{D}^i)$ is the prior distribution of current task $i$, then we have

$$-\mathbb{E}_{\mathcal{D}^i,z,x}\left[\log \frac{h(z,x)}{\sum_{\mathcal{D}^j \in \mathcal{D}^{\mathcal{T}}} h(z,x^j)}\right] \geq -\mathbb{E}_{x \sim \mathcal{D}^i}\left[D_{\mathrm{KL}}(g(\cdot|x)\|p(\mathcal{D}^i))\right] \qquad (9)$$

where $x^j$ are observations sampled from task $\mathcal{D}^j$ and $i \in \{1, 2, \cdots, N\}$.

*Proof.* We introduce the mutual information $\mathcal{I}(z, \mathcal{D}^i)$ between the learned skill $z$ and the current task's data distribution $\mathcal{D}^i$, and rewrite the inequality as below,

$$-\mathbb{E}_{\mathcal{D}^i,z,x}\left[\log \frac{h(z,x)}{\sum_{\mathcal{D}^* \in \mathcal{D}^{\mathcal{T}}} h(z,x^*)}\right] \geq \mathcal{I}(z;\mathcal{D}^i) \geq -\mathbb{E}_{x \sim \mathcal{D}^i}\left[D_{\mathrm{KL}}(g(\cdot|x)\|p(\mathcal{D}^i))\right] \qquad (18)$$

For the left side:

$$
\begin{aligned}
\mathcal{I}(z;\mathcal{D}^i) &= \mathbb{E}_{\mathcal{D}^i,z}\left[\log \frac{p(z|\mathcal{D}^i)}{p(z)}\right] \\
&\overset{(a)}{=} \mathbb{E}_{\mathcal{D}^i,z}\left[\log \mathbb{E}_x\left[\frac{g_\omega(z|x)}{p(z)}\right]\right] \\
&\geq \mathbb{E}_{\mathcal{D}^i,z,x}\left[\log \frac{g_\omega(z|x)}{p(z)}\right] \\
&= -\mathbb{E}_{\mathcal{D}^i,z,x}\left[\log \left(\frac{p(z)}{g_\omega(z|x)}N\right)\right] + \log N \\
&\geq -\mathbb{E}_{\mathcal{D}^i,z,x}\left[\log \left(1 + \frac{p(z)}{g_\omega(z|x)}(N-1)\right)\right] + \log N \\
&\geq -\mathbb{E}_{\mathcal{D}^i,z,x}\left[\log \left(1 + \frac{p(z)}{g_\omega(z|x)}(N-1)\underset{\mathcal{D}^* \in \mathcal{D}^{\mathcal{T}\backslash i}}{\mathbb{E}}\frac{g_\omega(z|x^*)}{p(z)}\right)\right] + \log N \\
&= -\mathbb{E}_{\mathcal{D}^i,z,x}\left[\log \left(1 + \frac{p(z)}{g_\omega(z|x)}\sum_{\mathcal{D}^* \in \mathcal{D}^{\mathcal{T}\backslash i}}\frac{g_\omega(z|x^*)}{p(z)}\right)\right] + \log N \\
&= \mathbb{E}_{\mathcal{D}^i,z,x}\left[\log \left(\frac{\frac{g_\omega(z|x)}{p(z)}}{\frac{g_\omega(z|x)}{p(x)} + \sum_{\mathcal{D}^* \in \mathcal{D}^{\mathcal{T}\backslash i}}\frac{g_\omega(z|x^*)}{p(z)}}\right)\right] + \log N \\
&\geq \mathbb{E}_{\mathcal{D}^i,z,x}\left[\log \left(\frac{\frac{g_\omega(z|x)}{p(z)}}{\sum_{\mathcal{D}^* \in \mathcal{D}^{\mathcal{T}}}\frac{g_\omega(z|x^*)}{p(z)}}\right)\right] \\
&= \mathbb{E}_{\mathcal{D}^i,z,x}\left[\log \frac{h(z,x)}{\sum_{\mathcal{D}^* \in \mathcal{D}^{\mathcal{T}}} h(z,x^*)}\right],
\end{aligned}
\qquad (19)
$$

where $N$ is the number of the task set $\mathcal{D}^{\mathcal{T}}$. Here, $(a)$ is derived from Lemma A.1.

For the right side:

$$
\begin{aligned}
\mathcal{I}(z; \mathcal{D}^i) &= \int \int g_\omega(z, x) \log \frac{g_\omega(z|x)}{g_\omega(z)} \mathrm{d}z \mathrm{d}x \\
&= \int \int g_\omega(z, x) \log g_\omega(z|x) \mathrm{d}z \mathrm{d}x - \int g_\omega(z) \log g_\omega(z) \mathrm{d}z \\
&\overset{(b)}{\leq} \int \int g_\omega(z, x) \log g_\omega(z|a) \mathrm{d}z \mathrm{d}x - \int g_\omega(z) \log p(\mathcal{D}^i) \mathrm{d}z \\
&= \int \int g_\omega(x) g_\omega(z|x) \log \frac{g_\omega(z|x)}{p(\mathcal{D}^i)} \mathrm{d}z \mathrm{d}x \\
&= \mathbb{E}_{x \sim \mathcal{D}^i} \left[ \int g_\omega(z|x) \log \frac{g_\omega(z|x)}{p(\mathcal{D}^i)} \mathrm{d}z \right] \\
&= \mathbb{E}_{x \sim \mathcal{D}^i} \left[ D_{\mathrm{KL}}(g_\omega(\cdot|x) \| p(\mathcal{D}^i)) \right]
\end{aligned}
\tag{20}
$$

The inequality at $(b)$ is derived from $D_{\mathrm{KL}}(g_\omega(\cdot) \| p(\mathcal{D}^i) \geq 0$. $\qquad \square$

## B  PSEUDOCODE FOR HISSD

---
**Algorithm 1** HiSSD for Offline Multi-Task MARL
---
1: **Inputs:**
2: High-Level Planner $\pi_{\theta_h}$, Low-Level Controller $\pi_{\theta_l}$, Value Net $V_\xi^{\mathrm{tot}}$ and $\bar{V}_{\bar{\xi}}^{\mathrm{tot}}$, Forward Predictor $f_\phi$, Task-Specifc Skill Encoder $g_\omega$, Training Steps $T$, Task Numbers $N$, Offline Multi-Task Dataset $\mathcal{D}^{\mathcal{T}}$, Current Training Task $i$ and its data $\mathcal{D}^i \sim \mathcal{D}^{\mathcal{T}}$, Agent Numbers $\{K^i\}_{i=1}^{\mathcal{T}}$, Batch Size $B$, learning rate $\delta$, target update rate $\tau$.
3: **Training:**
4: **for** each timestep $n$ in $1..N$ **do**
5: $\quad \mathcal{D}^i \sim \mathcal{D}^{\mathcal{T}}$ `# Sample one task data from the offline dataset.`
6: $\quad \{s_t, r_t, \{o_t, a_t, o_{t+1}\}_{k=1}^K\}_{j=1}^B \sim \mathcal{D}^i$
7: $\quad c_t^{1:k} \leftarrow \pi_{\theta_h}(o_t^{1:k})$ `# Infer common skills.`
8: $\quad$ Compute $V_\xi^{\mathrm{tot}}(o_t^{1:k})$, $\bar{V}_{\bar{\xi}}^{\mathrm{tot}}(o_{t+1}^{1:k})$, and $\bar{V}_{\bar{\xi}}^{\mathrm{tot}}(f_\phi(c_t^{1:k}))$ with $s_t$
9: $\quad s'_{t+1} \leftarrow f_\phi(c_t^{1:k})$
10: $\quad z_t^{1:k} \leftarrow g_\omega(o_t^{1:k})$ `# Infer task-specific skills.`
11: $\quad a_t'^{1:k} \leftarrow \pi_{\theta_l}(z_t^{1:k}, o_t^{1:k}, c_t^{1:k})$
12: $\quad$ Optimizing Eq. 5 with $V_\xi^{\mathrm{tot}}(o_t^{1:k})$, $\bar{V}_{\bar{\xi}}^{\mathrm{tot}}(o_{t+1}^{1:k})$, and $r_t$
13: $\quad$ Optimizing Eq. 7 with $V_\xi^{\mathrm{tot}}(o_t^{1:k})$, $\bar{V}_{\bar{\xi}}^{\mathrm{tot}}(f_\phi(c_t^{1:k}))$, $r_t$, and $s'_{t+1}$
14: $\quad$ Optimizing Eq. 11 with $c_t^{1:k}$, $z_t^{1:k}$, $a_t'^{1:k}$, and samples from other tasks
15: **end for**
16: **Execution:**
17: **for** each timestep $t$ in current environment **do**
18: $\quad c_t^{1:k} \leftarrow \pi_{\theta_h}(o_t^{1:k})$ `# Infer common skills.`
19: $\quad z_t \leftarrow g_\omega(o_t^{1:k})$ `# Infer task-specific skills.`
20: $\quad a_t^{1:k} \leftarrow \pi_{\theta_l}(z_t^{1:k}, o_t^{1:k}, c_t^{1:k})$
21: **end for**
---

## C  BENCHMARKS AND DATASETS

### C.1  SMAC

The StarCraft Multi-Agent Challenge (SMAC) (Samvelyan et al., 2019) is a widely used cooperative multi-agent testbed that contains diverse StarCraft micromanagement scenarios. In this paper, we utilize three distinct SMAC task sets defined by Zhang et al. (2023): *Marine-Hard*, *Marine-Easy*, and *Stalker-Zealot* to evaluate the capacity of transferring policy to unseen tasks. The Marine-Hard and Marine-Easy task sets include various marine battle scenarios, and the trained multi-agent policy needs to control groups of allied marines to confront equivalent or superior numbers of built-in-AI enemy marines. The stalker-zealot task set includes several tasks with symmetric stalkers and

Table 4: Descriptions of tasks in the Marine-Easy task set.

| Task type | Task | Ally units | Enemy units | Properties |
|---|---|---|---|---|
| Source tasks | 3m | 3 Marines | 3 Marines | homogeneous & symmetric |
| | 5m | 5 Marines | 5 Marines | homogeneous & symmetric |
| | 10m | 10 Marines | 10 Marines | homogeneous & symmetric |
| Unseen tasks | 4m | 4 Marines | 4 Marines | homogeneous & symmetric |
| | 6m | 6 Marines | 6 Marines | homogeneous & symmetric |
| | 7m | 7 Marines | 7 Marines | homogeneous & symmetric |
| | 8m | 8 Marines | 8 Marines | homogeneous & symmetric |
| | 9m | 9 Marines | 9 Marines | homogeneous & symmetric |
| | 11m | 11 Marines | 11 Marines | homogeneous & symmetric |
| | 12m | 12 Marines | 12 Marines | homogeneous & symmetric |

Table 5: Descriptions of tasks in the Marine-Hard task set.

| Task type | Task | Ally units | Enemy units | Properties |
|---|---|---|---|---|
| Source tasks | 3m | 3 Marines | 3 Marines | homogeneous & symmetric |
| | 5m_vs_6m | 5 Marines | 6 Marines | homogeneous & asymmetric |
| | 9m_vs_10m | 9 Marines | 10 Marines | homogeneous & asymmetric |
| Unseen tasks | 4m | 4 Marines | 4 Marines | homogeneous & symmetric |
| | 5m | 5 Marines | 5 Marines | homogeneous & symmetric |
| | 10m | 10 Marines | 10 Marines | homogeneous & symmetric |
| | 12m | 12 Marines | 12 Marines | homogeneous & symmetric |
| | 7m_vs_8m | 7 Marines | 8 Marines | homogeneous & asymmetric |
| | 8m_vs_9m | 8 Marines | 9 Marines | homogeneous & asymmetric |
| | 10m_vs_11m | 10 Marines | 11 Marines | homogeneous & asymmetric |
| | 10m_vs_12m | 10 Marines | 12 Marines | homogeneous & asymmetric |
| | 13m_vs_15m | 13 Marines | 15 Marines | homogeneous & asymmetric |

zealots on each side. To achieve generalization to unseen tasks with limited sources, we train on three selected tasks and reserve the remaining tasks for evaluation. Detailed attributes of these task sets are enumerated in Tables 4, 5, and 6.

As stated in the experiments section, we utilize the same offline multi-task dataset as Zhang et al. (2023) to maintain a fair comparison. Definitions of these four qualities are listed below:

- The **expert** dataset contains trajectory data collected by a QMIX policy trained with $2,000,000$ steps of environment interactions. The test win rate of the trained QMIX policy (as the expert policy) is recorded for constructing medium datasets.

- The **medium** dataset contains trajectory data collected by a QMIX policy (as the medium policy) whose test win rate is half of the expert QMIX policy.

- The **medium-expert** dataset mixes data from the expert and the medium dataset to acquire a more diverse dataset.

- The **medium-replay** dataset is the replay buffer of the medium policy, containing trajectory data with lower qualities.

The Properties of offline datasets with different qualities are detailed in Table 7.

## C.2  MAMuJoCo

Multi-Agent MuJoCo (Peng et al., 2021) is a benchmark developed for assessing and comparing the effectiveness of algorithms in continuous multi-agent robotic control. In MAMuJoCo, a robotic

Table 6: Descriptions of tasks in the Stalker-Zealot task set.

| Task type | Task | Ally units | Enemy units | Properties |
|---|---|---|---|---|
| Source tasks | 2s3z | 2 Stalkers, 3 Zealots | 2 Stalkers, 3 Zealots | heterogeneous & symmetric |
| | 2s4z | 2 Stalkers, 4 Zealots | 2 Stalkers, 4 Zealots | heterogeneous & symmetric |
| | 3s5z | 3 Stalkers, 5 Zealots | 3 Stalkers, 5 Zealots | heterogeneous & symmetric |
| Unseen tasks | 1s3z | 1 Stalkers, 3 Zealots | 1 Stalkers, 3 Zealots | heterogeneous & symmetric |
| | 1s4z | 1 Stalkers, 4 Zealots | 1 Stalkers, 4 Zealots | heterogeneous & symmetric |
| | 1s5z | 1 Stalkers, 5 Zealots | 1 Stalkers, 5 Zealots | heterogeneous & symmetric |
| | 2s5z | 2 Stalkers, 5 Zealots | 2 Stalkers, 5 Zealots | heterogeneous & symmetric |
| | 3s3z | 3 Stalkers, 3 Zealots | 3 Stalkers, 3 Zealots | heterogeneous & symmetric |
| | 3s4z | 3 Stalkers, 4 Zealots | 3 Stalkers, 4 Zealots | heterogeneous & symmetric |
| | 4s3z | 4 Stalkers, 3 Zealots | 4 Stalkers, 3 Zealots | heterogeneous & symmetric |
| | 4s4z | 4 Stalkers, 4 Zealots | 4 Stalkers, 4 Zealots | heterogeneous & symmetric |
| | 4s5z | 4 Stalkers, 5 Zealots | 4 Stalkers, 5 Zealots | heterogeneous & symmetric |

system is partitioned into independent agents, each tasked with controlling a specific set of joints to accomplish shared objectives. To conduct offline multi-task learning, we choose *HalfCheetah*-v2 as our base scenarios and partition the robotic system into six agents. Besides the complete HalfCheetah robot, we design six new tasks by disabling each agent. Therefore, our MAMuJoCo multi-task data comprises seven tasks. Each task's name corresponds to the joint controlled by the disabled agent. We follow Wang et al. (2023a) and generate offline data using the policy trained by HAPPO (Kuba et al., 2022). The hyperparameter `env_args.agent_obsk` (determines up to which connection distance agents will be able to form observations) is set to 1. We list the average return of our datasets in Table 8.

# D IMPLEMENTATION DETAILS

## D.1 DETAILS OF HiSSD IN SMAC

We follow prior works and decompose the observation to process the varying observation sizes among tasks. The local observation $o_i$ is decomposed into several portions, the one including agent $i$'s own information $o_i^{\text{own}}$, and the other contains other entities' information $\{o_{i,j}^{\text{entity}}\}$. We employ a transformer model for parallel tensor processing to embed these portions. Each portion is embedded by a separate fully connected layer with an output dimension of 64, and the transformer processes these embedding according to the attention mechanism,

$$Q = W^Q([o_i^{\text{own}}, o_{i,1}^{\text{entity}}, \dots]), K = W^K([o_i^{\text{own}}, o_{i,1}^{\text{entity}}, \dots]), V = W^V([o_i^{\text{own}}, o_{i,1}^{\text{entity}}, \dots]),$$

$$[e_i^{\text{own}}, e_{i,1}^{\text{entity}}, \dots] = \text{softmax}\left(\frac{QK^{\text{T}}}{\sqrt{d_K}}\right)V, \tag{21}$$

$$\text{where } [o_i^{\text{own}}, o_{i,1}^{\text{entity}}, \dots] = \text{Decompose}(o_i), \ \ d_K = \dim(K).$$

Table 7: Properties of offline datasets in SMAC with different qualities.

| Tasks | Quality | Trajectories | Average Return | Average Win Rate |
|-------|---------|--------------|----------------|------------------|
| 3m | expert | 2000 | 19.8929 | 0.9910 |
| | medium | 2000 | 13.9869 | 0.5402 |
| | medium-expert | 4000 | 16.9399 | 0.7656 |
| | medium-replay | 3603 | N/A | N/A |
| 5m | expert | 2000 | 19.9380 | 0.9937 |
| | medium | 2000 | 17.3288 | 0.7411 |
| | medium-expert | 4000 | 18.6334 | 0.8674 |
| | medium-replay | 711 | N/A | N/A |
| 10m | expert | 2000 | 19.9438 | 0.9922 |
| | medium | 2000 | 16.6297 | 0.5413 |
| | medium-expert | 4000 | 18.2595 | 0.7626 |
| | medium-replay | 571 | N/A | N/A |
| 5m_vs_6m | expert | 2000 | 17.3424 | 0.7185 |
| | medium | 2000 | 12.6408 | 0.2751 |
| | medium-expert | 4000 | 14.9916 | 0.4968 |
| | medium-replay | 32607 | N/A | N/A |
| 9m_vs_10m | expert | 2000 | 19.6140 | 0.9431 |
| | medium | 2000 | 15.5049 | 0.4146 |
| | medium-expert | 4000 | 17.5594 | 0.6789 |
| | medium-replay | 13731 | N/A | N/A |
| 2s3z | expert | 2000 | 19.7655 | 0.9602 |
| | medium | 2000 | 16.6279 | 0.4465 |
| | medium-expert | 4000 | 18.1967 | 0.7034 |
| | medium-replay | 4505 | N/A | N/A |
| 2s4z | expert | 2000 | 19.7402 | 0.9509 |
| | medium | 2000 | 16.8735 | 0.4965 |
| | medium-expert | 4000 | 18.3069 | 0.7237 |
| | medium-replay | 6172 | N/A | N/A |
| 3s5z | expert | 2000 | 19.7850 | 0.9518 |
| | medium | 2000 | 16.3126 | 0.3114 |
| | medium-expert | 4000 | 18.0488 | 0.6316 |
| | medium-replay | 11528 | N/A | N/A |

Table 8: Properties of offline datasets on HalfCheetah-v2 in MAMuJoCo.

| Tasks | Trajectories | Average Return |
|-------|--------------|----------------|
| complete | 100 | 2881.63 |
| back thigh | 100 | 2764.65 |
| back foot | 100 | 2880.78 |
| front thigh | 100 | 2011.00 |
| front shin | 100 | 3048.01 |

The common skill encoder, the task-specific skill encoder, the individual value network, and the action decoder use single-layer transformers (64-unit hidden layers) to process the decomposed observation. The mixing value network implements an attention block, following Zhang et al. (2023). The forward predictor comprises two single-layer transformers. To tackle the partial observability, we append the history hidden state $h_{t-1}^i$ in each agent's common skill encoder, value net, and action decoder when applying self-attention and thus get the output of $h_t^i$.

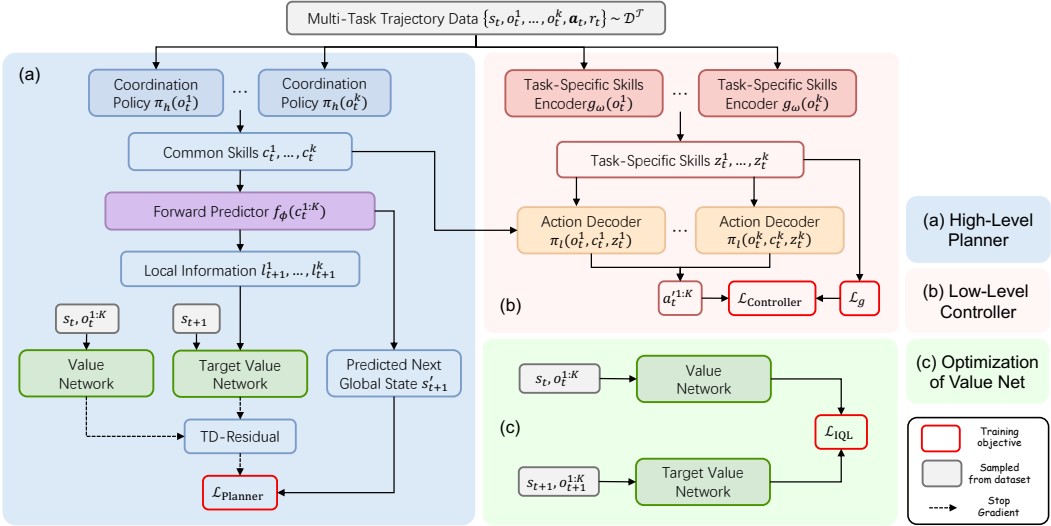

Figure 4: Overall framework of Hi-SSD in SMAC. HiSSD utilizes a hierarchical framework that jointly learns common and task-specific skills from offline multi-task data to improve multi-agent policy transfer. (a) The high-level planner with common skills. HiSSD integrates cooperative temporal knowledge into common skills and enables an offline exploration. (b) The low-level controller with task-specific skills. The task-specific knowledge can guide the action execution adaptively among tasks. (c) HiSSD uses an implicit Q-learning objective to train the value network.

The common skills, extracted through the common skill encoder, are fed into the forward predictor for global state prediction and the action decoder for execution. Our framework employs a dual-transformer architecture for global state prediction: the first transformer integrates enemy information, while the second processes ally and preprocessed enemy information to predict subsequent global states. For the action execution, we concatenate the decomposed information outputted by the action decoder with the task-specific skills and feed them into an MLP to get the real actions. Figure 4 illustrates the overall framework of HiSSD in SMAC. The hyperparameters used in SMAC are listed in Table 9.

Table 9: Hyperparameters of HiSSD for offline multi-task SMAC.

| Hyperparameter | Setting |
|---|---|
| Hidden layer dimension | 64 |
| Hidden units in MLP | 128 |
| Attention dimension | 64 |
| Skill dimension per token | 64 |
| Discount factor $\gamma$ | 0.99 |
| $\alpha$ | 10 |
| $\beta$ | 0.05 |
| $\epsilon$ | 0.9 |
| Trajectories per batch | 32 |
| Training steps | 30000 |
| Optimizer | Adam |
| Learning rate | 0.0001 |
| Weight decay | 0.001 in *Stalker-Zealot* 0.0001 in others |

## D.2    DETAILS OF HiSSD IN MAMUJOCO

The observation shapes are consistent among tasks and we do not apply the observation decomposition deployed in SMAC. The common skill encoder, the task-specific skill encoder, the individual value network, and the action decoder use single-layer transformers (64-unit hidden layers) to process the observation. The common skills, extracted through the common skill encoder, are fed into the forward predictor for global state prediction and the action decoder for execution. We concatenate the decomposed information outputted by the action decoder with the task-specific skills and feed them into an MLP to get the real actions. The hyperparameters used in MAMuJoCo are listed in Table 10.

Table 10: Hyperparameters of HiSSD for offline multi-task MAMuJoCo.

| Hyperparameter | Setting |
|---|---|
| Hidden layer dimension | 256 |
| Hidden units in MLP | 256 |
| Attention dimension | 256 |
| Skill dimension per token | 256 |
| Discount factor $\gamma$ | 0.99 |
| Target update rate | 0.005 |
| $\alpha$ | 10.0 |
| $\beta$ | 2.0 |
| $\epsilon$ | 0.9 |
| Batch size | 128 |
| Training steps | 1000000 |
| Optimizer | Adam |
| Learning rate | 0.0005 |

## D.3    TRAINING COSTS

The training process of HiSSD with an NVIDIA GeForce RTX 3090 GPU and a 32-core CPU costs 12-14 hours typically. Our released implementation of HiSSD follows Apache License 2.0, the same as the PyMARL framework.

## E    ADDITIONAL RESULTS

### E.1    RESULTS ON OTHER TASK SET IN SMAC.

We follow the multi-task learning settings in ODIS (Zhang et al., 2023) and conduct additional experiments on two offline multi-task task sets: *Marine-Easy* and *Stalker-Zealot*. The results are presented in Tables 11 and 12. We also present the average win rates on *Stalker-Zealot* task set in Table 13 to show the improvement of HiSSD on tasks with heterogeneous units clearly. For the *Marine-Easy* task set, HiSSD gains convincing performance in most source and unseen tasks compared to other baselines. For the *Stalker-Zealot* task set, HiSSD obtains competitive performance and surpasses ODIS when the dataset is generated by near-optimal policy (i.e., *Expert* and *Medium-Expert*). Moreover, both skill-learning-based methods fail to outperform BC-based methods on some tasks in *Stalker-Zealot* task set. We suspect this is due to the different task properties between *Marine* and *Stalker-Zealot* task set as the controlled items in *Marine* are homogeneous and in *Stalker-Zealot* are heterogeneous. Therefore, it is more difficult for the policy to learn generalizable cooperative patterns from the limited dataset in the *Stalker-Zealot* task set.

### E.2    SENSITIVITY ANALYSIS OF HYPER-PARAMETERS.

To investigate the model's sensitivity analysis to $\epsilon$ in Eq. 5, $\alpha$ in Eq. 7, and $\beta$ in Eq. 11, we conduct experiments on Marine-Hard task set in SMAC with expert data qualities. We list the results in Tables 14, 15, and 16. While our framework incorporates additional architectural components and

Table 11: Average test win rates in the *Marine-Easy* task set with different data qualities. Results of BC-best stands for the best test win rates between BC-t and BC-r.

| Tasks | Expert | | | | Medium | | | |
|---|---|---|---|---|---|---|---|---|
| | BC-best | UPDeT-m | ODIS | Hi-SSD (Ours) | BC-best | UPDeT-m | ODIS | Hi-SSD (Ours) |
| | | | | Source Tasks | | | | |
| 3m | 94.5± 4.6 | 83.6±12.6 | 97.7± 2.6 | **99.5± 8.1** | 67.2± 4.7 | 60.2±29.9 | 57.8± 9.2 | **74.7±14.6** |
| 5m | 94.4± 7.6 | 74.8±22.9 | 95.3± 5.2 | **99.9± 0.0** | 79.2± 5.9 | 67.8± 5.9 | **82.8± 5.2** | 81.6±10.8 |
| 10m | 86.1±22.7 | 83.6±19.2 | 88.3±20.3 | **95.2± 8.4** | 63.1± 7.2 | 48.8± 7.9 | 71.9± 6.6 | **84.8± 8.6** |
| | | | | Unseen Tasks | | | | |
| 4m | 91.2± 1.6 | 53.0±32.3 | 90.6± 7.0 | **94.4± 2.9** | 62.5±11.6 | 41.7±17.4 | 63.3±16.1 | **74.5±15.5** |
| 6m | 75.3±22.6 | 37.9± 8.6 | 79.7±17.5 | **99.7± 0.3** | 86.0± 4.7 | 75.8±22.7 | **89.8±17.6** | 88.0±10.0 |
| 7m | 70.3±11.0 | 44.2±13.2 | 72.7±16.9 | **99.1± 0.7** | **99.9± 0.0** | 65.2±25.2 | 96.1± 1.4 | 97.3± 2.3 |
| 8m | 74.7±16.5 | 51.7±26.2 | 80.9±14.4 | **99.8± 0.3** | 96.9± 2.2 | 88.4±13.7 | **97.7± 2.6** | 93.8± 5.2 |
| 9m | 97.7± 2.6 | 76.3±13.4 | 99.2± 1.4 | **99.9± 0.0** | 78.9±11.8 | 64.8±35.6 | **87.5± 2.2** | 75.2±15.5 |
| 11m | 83.3±11.8 | 53.6±22.4 | 83.6±12.4 | **99.2± 0.8** | 42.2± 4.7 | 23.4±11.8 | **64.7± 3.1** | 62.0±21.8 |
| 12m | 56.7±30.0 | 44.3±22.8 | 70.3±30.2 | **99.7± 1.1** | 29.7±23.4 | 13.5±11.7 | 41.4± 6.0 | **55.5±25.7** |
| | Medium-Expert | | | | Medium-Replay | | | |
| | | | | Source Tasks | | | | |
| 3m | 81.3±18.8 | 48.4±36.8 | 89.8± 9.7 | **90.9± 5.9** | 77.8± 3.2 | 29.7±10.0 | 79.7± 4.7 | **87.7± 2.9** |
| 5m | 74.0± 2.9 | 64.1±17.9 | **83.7±16.0** | 79.4± 6.9 | 5.5± 5.6 | 6.2±10.8 | 3.1± 5.4 | **87.5± 2.9** |
| 10m | 90.6± 3.1 | 68.8±23.8 | **93.8± 4.4** | 60.2±21.1 | 0.0± 0.0 | 0.0± 0.0 | 0.0± 0.0 | **84.2±4.9** |
| | | | | Unseen Tasks | | | | |
| 4m | 35.2±38.0 | 43.7±25.0 | 57.8±18.8 | **70.9± 9.1** | 67.2± 4.7 | 25.0±22.6 | 25.0± 5.4 | **71.6± 4.1** |
| 6m | 42.2± 1.6 | 47.7±30.0 | **76.0± 6.0** | 70.6± 6.1 | 7.8±10.2 | 0.0± 0.0 | 3.1± 5.4 | **99.8± 0.3** |
| 7m | 65.6±16.4 | 57.8±32.9 | 66.4±14.6 | **85.0±11.7** | 0.8± 1.4 | 0.0± 0.0 | 0.0± 0.0 | **99.8± 0.3** |
| 8m | 40.3±42.6 | 40.6±19.3 | 43.8±11.5 | **72.8± 9.5** | 0.8± 1.4 | 0.0± 0.0 | 1.6± 1.6 | **96.7± 0.3** |
| 9m | 70.8±16.6 | 47.7±24.8 | 73.4±16.2 | **80.0±14.6** | 0.8± 1.4 | 0.0± 0.0 | 0.0± 0.0 | **88.8± 1.3** |
| 11m | 55.5±12.4 | **85.9±14.2** | 68.8±20.3 | 70.9± 5.9 | 0.0± 0.0 | 0.0± 0.0 | 0.0± 0.0 | **45.6± 4.5** |
| 12m | 29.7±29.8 | 46.1±15.5 | 62.5± 8.0 | **62.7± 7.8** | 0.0± 0.0 | 0.0± 0.0 | 0.0± 0.0 | **38.0± 3.7** |

learning objectives compared to prior approaches, empirical evaluations demonstrate its consistent robustness across varying hyperparameter configurations.

### E.3 COMPARED TO HYGEN

HyGen (Zhang et al., 2024) is a recent work that focuses on integrating offline pertaining and online exploration to speed up multi-task MARL for policy transfer. HyGen first pre-trains the skill space with an action decoder using the global information, then implements online exploration to maintain a hybrid replay buffer using offline and online data, improving the high-level policy and refining the action decoder simultaneously. Compared to HyGen, our method requires no online exploration and pretraining steps. HyGen only conducts common skills learning while our method leverages task-specific skills to complement the common skills discovery. We compare our method to HyGen in the SMAC's Marine hard task set and propose the empirical results in Table 17. While HiSSD demonstrates superior performance over HyGen across most tasks under expert-level data quality conditions, HyGen capitalizes on online exploration to achieve policy refinement when provided with moderate-quality offline datasets.

### E.4 EXTENDED ABLATION STUDY

To demonstrate the effectiveness of learning task-specific skills, we conduct a variant of HiSSD named *w/o Specific* and present the empirical results in Table 18. We find that policy with only common skills significantly outperforms the BC-based method, learning both skills further improves the transfer capability of the policy.

Table 12: Average test win rates in the *Stalker-Zealot* task set with different data qualities. **Bold** represents the best score in each task.

| Tasks | Expert | | | | Medium | | | |
|---|---|---|---|---|---|---|---|---|
| | BC-best | UPDeT-m | ODIS | Hi-SSD (Ours) | BC-best | UPDeT-m | ODIS | Hi-SSD (Ours) |
| Source Tasks | | | | | | | | |
| 2s3z | 93.1± 4.6 | 50.0±33.4 | **97.7± 2.6** | 95.2± 1.0 | 48.8± 9.8 | 35.0±23.0 | **49.2± 8.4** | 32.3±11.7 |
| 2s4z | 78.1± 8.1 | 23.4±26.6 | 60.9± 6.8 | **79.8± 6.0** | 12.5± 8.1 | 18.8±10.3 | **32.8±12.2** | 17.0± 2.2 |
| 3s5z | 92.5±4.2 | 17.2±19.8 | 87.5± 9.6 | **92.8± 5.0** | 24.4±12.4 | 25.6±24.2 | **28.9± 6.8** | 24.4± 7.9 |
| Unseen Tasks | | | | | | | | |
| 1s3z | 45.6±23.8 | 1.6± 1.6 | 76.6± 3.5 | **81.6±15.2** | 21.9±37.6 | 3.8± 5.0 | 41.4±18.8 | **44.2± 9.9** |
| 1s4z | **60.0±32.3** | 26.6±19.3 | 17.2±10.5 | 42.0±26.1 | 6.2± 7.7 | 2.5± 3.6 | **50.7± 7.5** | 18.1±11.0 |
| 1s5z | **45.6±26.9** | 29.7±26.4 | 2.5 ± 2.3 | 16.7±12.3 | 3.1± 2.6 | 5.0± 4.2 | **14.1± 8.4** | 2.5± 2.2 |
| 2s5z | 75.6±11.9 | 23.4±22.2 | 27.3± 6.0 | **79.7± 2.2** | 14.4± 9.0 | 16.9±14.1 | **32.0± 4.6** | 11.3± 3.7 |
| 3s3z | 80.6± 9.1 | 20.3±10.9 | **89.1± 5.2** | 88.0± 4.5 | **45.6±14.6** | 24.4±28.6 | 23.4± 9.2 | 21.9±10.7 |
| 3s4z | 92.5± 5.1 | 12.5±19.9 | **96.9± 2.2** | 88.1± 9.0 | 40.0±19.0 | 28.8±31.6 | **50.8±15.5** | 17.2± 4.5 |
| 4s3z | 67.5±19.8 | 6.2± 4.4 | 64.1±13.0 | **88.6± 4.1** | 28.8±26.4 | 11.2±18.0 | 13.3± 7.5 | **31.9±23.2** |
| 4s4z | 53.1±18.4 | 7.8±13.5 | **79.7±10.9** | 73.4± 5.2 | **20.0±12.0** | 1.2± 1.5 | 12.5± 7.0 | 13.2± 6.5 |
| 4s5z | 40.6±19.1 | 5.5± 7.8 | **86.7±12.6** | 65.6± 3.7 | **14.4± 8.5** | 5.6± 8.5 | 7.0± 4.1 | 4.5± 1.3 |
| 4s6z | 48.1±23.8 | 4.7± 6.4 | **88.3± 8.4** | 68.4± 4.9 | **3.8± 3.6** | 1.9± 2.5 | 1.6± 1.6 | 0.9± 0.9 |
| | Medium-Expert | | | | Medium-Replay | | | |
| Source Tasks | | | | | | | | |
| 2s3z | 57.5±25.1 | 57.5±27.1 | 58.6±15.5 | **68.1± 8.1** | 3.1± 2.6 | 14.4±13.2 | **15.6±18.2** | 9.0± 1.5 |
| 2s4z | 37.5±15.3 | **53.1±24.6** | 41.4± 7.8 | 41.9±10.2 | 5.2± 7.4 | **12.5± 9.7** | 7.8± 5.2 | 6.0± 1.2 |
| 3s5z | **63.1±13.3** | 35.0±23.5 | 41.4±18.5 | 57.8±10.7 | 31.3± 6.3 | **20.0±16.6** | 18.8± 3.1 | 17.5± 2.0 |
| Unseen Tasks | | | | | | | | |
| 1s3z | 55.6±37.7 | 4.4± 8.8 | 72.7±12.2 | **73.0±10.2** | 24.0±15.4 | 0.0± 0.0 | 21.1±20.4 | **36.3± 7.1** |
| 1s4z | 25.0±30.7 | 11.9± 9.8 | **44.5±20.3** | 32.3±30.5 | 2.1± 2.9 | 7.5±10.0 | 6.2± 7.7 | **24.8± 9.1** |
| 1s5z | 14.4±19.4 | 3.8± 4.6 | **42.2±31.4** | 9.4± 9.5 | 7.3± 6.4 | **11.9± 9.6** | 7.8± 6.4 | 4.4± 2.2 |
| 2s5z | 26.9±20.2 | 37.5±22.5 | **43.0±10.7** | 25.6± 7.8 | 12.5±15.5 | **20.0±16.8** | 14.1± 8.1 | 16.5± 2.8 |
| 3s3z | 35.6±18.0 | 33.8±15.0 | 50.0±13.3 | **56.6±25.6** | **35.4±12.1** | 17.5±12.3 | 25.0±20.1 | 9.6± 3.3 |
| 3s4z | **74.4±16.3** | 43.1±20.7 | 52.3± 9.5 | 71.7± 9.7 | 20.8± 9.0 | 15.6±11.2 | 19.5±16.6 | **22.5±10.6** |
| 4s3z | **69.8± 7.8** | 23.8±21.0 | 17.2± 7.2 | 60.5±15.1 | **17.7± 5.3** | 11.2±15.0 | 8.6±14.9 | 11.0±10.4 |
| 4s4z | **41.9±14.9** | 10.6±13.8 | 20.3± 6.8 | 37.3± 9.4 | **15.6± 6.8** | 5.6± 9.8 | 4.7± 8.1 | 9.4± 1.8 |
| 4s5z | 17.3± 5.3 | 11.9±16.1 | **21.9± 2.2** | 17.0± 4.1 | 1.0± 1.5 | **10.6±19.7** | 0.8± 1.4 | 0.8± 0.8 |
| 4s6z | 13.8± 3.2 | 5.0± 8.5 | 18.0± 5.1 | **19.7± 5.9** | 0.0± 0.0 | **6.9±13.8** | 2.3± 4.1 | 0.4± 0.3 |

Table 13: Average test win rates in the *Stalker-Zealot* task set with different data qualities.

| Data Qualities | BC-best | UPDeT-m | ODIS | HiSSD |
|---|---|---|---|---|
| Source Tasks | | | | |
| Expert | 87.9± 5.6 | 30.2±26.6 | 82.0± 6.3 | **89.3± 4.0** |
| Medium | 28.6±10.1 | 26.5±19.2 | **37.0± 9.1** | 24.6± 7.3 |
| Medium-Expert | 52.9±17.9 | 48.5±25.1 | 47.1±13.9 | **55.9± 9.7** |
| Medium-Replay | 13.2± 5.4 | **15.6±13.2** | 14.1± 8.8 | 10.8± 1.6 |
| Target Tasks | | | | |
| Expert | 60.9±19.0 | 13.8±13.2 | 62.8± 7.5 | **69.2± 8.7** |
| Medium | 19.8±14.1 | 10.1±11.8 | **24.7± 8.4** | 16.6± 7.4 |
| Medium-Expert | 37.5±17.4 | 18.6±14.1 | 38.2±11.9 | **40.3±12.8** |
| Medium-Replay | 13.6±7.6 | 10.7±11.8 | 11.0±10.8 | **13.6± 4.8** |

Table 14: Average test win rates on *Marine-Hard* task set in SMAC with expert data qualities for the model's sensitivity analysis to $\epsilon$. $\epsilon$ balances the imitation and exploration in offline value estimation.

| Tasks | Expert | | | | |
|---|---|---|---|---|---|
| | $\epsilon = 0.1$ | $\epsilon = 0.3$ | $\epsilon = 0.5$ | $\epsilon = 0.7$ | $\epsilon = 0.9$ (**Ours**) |
| Source Tasks | | | | | |
| 3m | 99.6± 0.6 | **99.9± 0.0** | 99.4± 0.5 | 99.4± 0.5 | 99.5± 0.3 |
| 5m6m | 73.5± 5.1 | 71.3± 3.3 | 70.8± 2.4 | **74.4± 1.5** | 66.1± 7.0 |
| 9m10m | 98.3± 0.3 | 97.1± 0.3 | **99.0± 0.3** | 96.7± 0.6 | 95.5± 2.7 |
| Unseen Tasks | | | | | |
| 4m | 97.7± 1.6 | 99.0± 0.3 | 99.0± 0.8 | 98.4± 0.5 | **99.2± 1.2** |
| 5m | **99.8± 0.3** | **99.8± 0.3** | 99.2± 0.3 | 99.1± 1.2 | 99.2± 1.2 |
| 10m | 99.2± 0.8 | 98.5± 1.6 | 99.2± 0.3 | **99.2± 0.3** | 98.4± 0.8 |
| 12m | 69.0±33.7 | 67.9±17.0 | 71.0±26.0 | **87.5± 7.6** | 75.5±17.9 |
| 7m8m | 29.2± 2.8 | 23.1± 4.9 | 32.7± 3.8 | 25.4± 2.9 | **35.3± 9.8** |
| 8m9m | 46.0±16.6 | 47.5± 9.2 | 45.4±14.8 | 44.6±11.4 | **47.0± 6.2** |
| 10m11m | 82.1± 4.3 | 81.4± 9.7 | 80.4± 3.1 | 72.3±13.8 | **86.3±14.6** |
| 10m12m | 11.5± 6.9 | 9.4± 5.9 | 8.1± 2.2 | 7.3± 5.5 | **14.5± 9.1** |
| 13m15m | 0.2± 0.3 | 0.4± 0.6 | 0.6± 0.5 | 0.4± 0.6 | **1.3± 2.5** |

Table 15: Average test win rates on *Marine-Hard* task set in SMAC with expert data qualities for the model's sensitivity analysis to $\alpha$. $\alpha$ weights the TD-residual Eq. 7.

| Tasks | Expert | | | |
|---|---|---|---|---|
| | $\alpha = 1$ | $\alpha = 5$ | $\alpha = 20$ | $\alpha = 10$ (**Ours**) |
| Source Tasks | | | | |
| 3m | 99.2± 0.6 | **99.9± 0.0** | 99.8± 0.3 | 99.5± 0.3 |
| 5m6m | 70.8± 4.7 | **71.0± 2.9** | 66.5± 1.1 | 66.1± 7.0 |
| 9m10m | 97.1± 1.8 | **97.9± 2.1** | 93.4± 1.6 | 95.5± 2.7 |
| Unseen Tasks | | | | |
| 4m | 98.4± 0.0 | 98.1± 0.5 | 99.0± 0.6 | **99.2± 1.2** |
| 5m | **99.8± 0.3** | **99.8± 0.3** | 98.8± 1.8 | 99.2± 1.2 |
| 10m | 99.4± 0.5 | 99.4± 0.9 | 99.0± 0.3 | **98.4± 0.8** |
| 12m | 59.2±36.2 | 63.8±33.6 | 66.5±26.5 | **75.5±17.9** |
| 7m8m | 34.5± 5.0 | 34.8±10.3 | 34.6± 2.8 | **35.3± 9.8** |
| 8m9m | 46.3± 2.5 | **59.2±13.3** | 53.3± 3.6 | 47.0± 6.2 |
| 10m11m | 83.3± 8.2 | 77.9±13.7 | 77.3± 6.8 | **86.3±14.6** |
| 10m12m | 10.2± 4.1 | 8.3± 8.3 | 11.9± 6.7 | **14.5± 9.1** |
| 13m15m | **2.5± 3.5** | 0.2± 0.3 | 1.3± 1.8 | 1.3± 2.5 |

Table 16: Average test win rates on *Marine-Hard* task set in SMAC with expert data qualities for the model's sensitivity analysis to $\beta$. $\beta$ controls the margin of regularization in Eq. 11.

| Tasks | Expert | | | | |
|---|---|---|---|---|---|
| | $\beta = 0.001$ | $\beta = 0.01$ | $\beta = 0.1$ | $\beta = 1$ | $\beta = 0.05$ (**Ours**) |
| Source Tasks | | | | | |
| 3m | 99.8± 0.3 | **99.9± 0.0** | 96.8± 4.5 | 99.5± 0.2 | 99.5± 0.3 |
| 5m6m | 65.3± 1.1 | 64.7± 2.4 | 64.5± 4.9 | 65.9± 2.7 | **66.1± 7.0** |
| 9m10m | 95.4± 2.6 | 95.3± 0.3 | 94.3± 0.6 | 92.0± 2.4 | **95.5± 2.7** |
| Unseen Tasks | | | | | |
| 4m | 98.3± 1.9 | 97.7± 0.6 | 97.5± 2.2 | 98.8± 0.3 | **99.2± 1.2** |
| 5m | 99.0± 0.8 | 98.8± 0.3 | 99.0± 0.9 | 99.1± 0.8 | **99.2± 1.2** |
| 10m | 97.5± 1.4 | 98.6± 0.3 | 96.7± 3.4 | 97.3± 1.3 | **98.4± 0.8** |
| 12m | 64.8±14.9 | 66.9±22.0 | 68.1±30.3 | 75.2±20.3 | **75.5±17.9** |
| 7m8m | 34.0± 5.5 | 35.4±12.0 | 36.5±13.0 | **40.6± 5.3** | 35.3± 9.8 |
| 8m9m | 50.8±13.3 | 40.2± 2.8 | **58.1±13.0** | 41.3± 7.2 | 47.0± 6.2 |
| 10m11m | 79.4± 2.3 | 79.6± 7.4 | 80.6±17.4 | 76.7±15.8 | **86.3±14.6** |
| 10m12m | 7.7± 1.6 | 11.2±10.6 | 7.9± 3.1 | 14.4± 8.5 | **14.5± 9.1** |
| 13m15m | 1.0± 0.8 | 1.7± 1.6 | 1.0± 1.1 | **2.1± 1.8** | 1.3± 2.5 |

Table 17: Average test win rates of the best policies over five random seeds in the task set *Marine-Hard* with *Expert* and *Medium* data qualities. Results of BC-best stands for the best test win rates between BC-t and BC-r.

| Tasks | Expert | | | Medium | | |
|---|---|---|---|---|---|---|
| | BC-best | HyGen | HiSSD (Ours) | BC-best | HyGen | HiSSD (Ours) |
| Source Tasks | | | | | | |
| 3m | 97.7± 2.6 | 99.1± 1.0 | **99.5± 0.3** | 65.4±14.7 | **91.5±11.0** | 62.7± 5.7 |
| 5m6m | 50.4± 2.3 | 61.2± 8.0 | **66.1± 7.0** | 21.9± 3.4 | **31.6± 7.0** | 26.4± 3.8 |
| 9m10m | 95.3± 1.6 | **96.4± 3.0** | 95.5± 2.7 | 63.8±10.9 | **79.2± 4.0** | 73.9± 2.3 |
| Unseen Tasks | | | | | | |
| 4m | 92.1± 3.5 | 95.8± 4.0 | **99.2± 1.2** | 48.8±21.1 | **91.4± 8.0** | 77.3±10.2 |
| 5m | 87.1±10.5 | 99.5± 1.0 | **99.2± 1.2** | 76.6±14.1 | **96.5± 6.0** | 88.4± 8.4 |
| 10m | 90.5± 3.8 | 93.5± 5.0 | **98.4± 0.8** | 56.2±20.6 | 96.4± 3.0 | **98.0± 0.3** |
| 12m | 70.8±15.2 | **85.2± 6.0** | 75.5±19.7 | 24.0±10.5 | 81.5±14.0 | **86.4± 6.0** |
| 7m8m | 18.8± 3.1 | 28.9±12.0 | **35.3± 9.8** | 1.6± 1.6 | **24.5± 9.0** | 14.2±10.1 |
| 8m9m | 15.8± 3.3 | 25.7± 9.0 | **47.0± 6.2** | 3.1± 3.8 | **24.5± 9.0** | 15.3± 2.8 |
| 10m11m | 45.3±11.1 | 57.2±13.0 | **86.3±14.6** | 19.7± 8.9 | 47.2±13.0 | 43.6± 4.6 |
| 10m12m | 1.0± 1.5 | 13.8± 4.0 | **14.5± 9.1** | 0.0± 0.0 | **5.2± 2.0** | 0.6± 0.5 |
| 13m15m | 0.0± 0.0 | **9.5± 5.0** | 1.3± 2.5 | 0.6± 1.3 | **9.3± 6.0** | 1.4± 2.4 |

Table 18: Additional ablation studies on HiSSD. We report average test win rates of the best policies over five random seeds in the task set *Marine-Hard* with different data qualities.

| Data Qualities | w/o Specific | BC-best | HiSSD |
|---|---|---|---|
| Source Tasks | | | |
| Expert | **85.9±16.4** | 81.1±21.8 | 84.5±17.1 |
| Medium | 53.1±20.0 | 50.3±20.1 | **55.9±19.2** |
| Medium-Expert | 63.5±26.1 | 41.7±18.5 | **64.2±23.5** |
| Medium-Replay | 47.7±25.1 | 46.5±26.7 | **51.9±24.0** |
| Target Tasks | | | |
| Expert | 55.0±37.4 | 46.8±36.8 | **61.2±37.4** |
| Medium | 37.6±34.2 | 25.6±26.8 | **48.6±40.1** |
| Medium-Expert | 51.2±34.3 | 38.8±33.0 | **57.6±37.5** |
| Medium-Replay | 42.8±36.2 | 41.2±33.7 | **48.7±39.0** |

