# OpenReview forum: "Learning Generalizable Skills from Offline Multi-Task Data for Multi-Agent Cooperation"
_ICLR.cc/2025/Conference — ICLR 2025 Poster_

### Official Review · Reviewer_sVE6 · 2024-11-01

**Soundness:** 2
**Presentation:** 2
**Contribution:** 3
**Rating:** 6
**Confidence:** 3

**Summary:**

This paper proposes an algorithm for multi-task multi-agent skill discovery from offline data, which is a quite meaningful and challenging research direction. The novelty (compared with ODIS) mainly lies in jointly learning common and task-specific skills.

**Strengths:**

(a) This paper targets at a challenging problem setup, i.e., recovering generalizable multi-agent skills from multi-task offline data.

(b) The evaluation results are sufficient and show consistent improvements over the SOTA algorithm -- ODIS.

**Weaknesses:**

(a) The algorithm framework, including both the multiple components to learn (Fig. 1) and the objective design (Eqs. 4, 7, 11), is quite convoluted compared to ODIS, which could potentially bring training difficulty.

(b) The writing needs to be significantly improved, especially for methodology part in Section 3.

(c) It's not clear why the common skills can integrate cooperative temporal knowledge and enable an offline exploration.

(d) Since all tasks share the same space for the local skills z^{1:K}_t, why z can embed task-specific knowledge?

(e) Multi-agent skills should provide abstractions at both temporal-level and subgroup-level. That is, each multi-agent skill should embed a (short) control sequence of a subgroup of agents. How is this aspect shown in the algorithm design?

(f) Thanks for releasing the code package. But it seems that the authors only made minimal changes to the readme file provided by ODIS. Please provide a detailed instruction (in the readme file) on how to reproduce all the reported results in this paper. Also, the released yaml files do not include the ones for MAMuJoCo.

**Questions:**

Please see the weakness part.

---

> ### Author Response · Authors · 2024-11-21
> **Reply to reviewer sVE6**
>
> Thank you for your time and feedback on HiSSD. We address each of your questions below.
>
> ### Weaknesses:
>
> **W1:** The framework is quite convoluted compared to previous methods, which could potentially bring training difficulty.
>
> **A:** In this paper, the training pipeline and hyperparameters tuning follow previous methods. To better illustrate the model sensitivity of our method, we conduct additional ablation studies for sensitive analysis and present the results below. The empirical results indicate that our method can be trained under different hyperparameter settings and no obvious training difficulty is observed.
>
> ***Sensitivity Analysis on $\epsilon$***
> *Source Tasks*
>
> | Tasks | $\epsilon=0.1$ | $\epsilon=0.3$  | $\epsilon=0.5$ | $\epsilon=0.7$ | $\epsilon=0.9$ (ours) |
> | ------- | ---------------- | ----------------- | ---------------- | ---------------- | ----------------------- |
> | 3m    | 99.6 ± 0.6    | **99.9 ± 0.0** | 99.4 ± 0.5    | 99.4 ± 0.5    | 99.5 ± 0.3           |
> | 5m6m  | 73.5± 5.1     | 71.3± 3.3      | 70.8± 2.4     | **74.4± 1.5** | 66.1± 7.0            |
> | 9m10m | 98.3± 0.3     | 97.1± 0.3      | **99.0± 0.3** | 96.7± 0.6     | 95.5± 2.7            |
>
> *Target Tasks*
>
> | Tasks  | $\epsilon=0.1$ | $\epsilon=0.3$ | $\epsilon=0.5$ | $\epsilon=0.7$ | $\epsilon=0.9$ (ours) |
> | -------- | ---------------- | ---------------- | ---------------- | ---------------- | ----------------------- |
> | 4m     | 97.7± 1.6     | 99.0± 0.3     | 99.0± 0.8     | 98.4± 0.5     | **99.2± 1.2**        |
> | 5m     | **99.8± 0.3** | **99.8± 0.3** | 99.2± 0.3     | 99.1± 1.2     | 99.2± 1.2            |
> | 10m    | 99.2± 0.8     | 98.5± 1.6     | 99.2± 0.3     | **99.2± 0.3** | 98.4± 0.8            |
> | 12m    | 69.0±33.7     | 67.9±17.0     | 71.0±26.0     | **87.5± 7.6** | 75.5±17.9            |
> | 7m8m   | 29.2± 2.8     | 23.1± 4.9     | 32.7± 3.8     | 25.4± 2.9     | **35.3± 9.8**        |
> | 8m9m   | 46.0±16.6     | **47.5± 9.2** | 45.4±14.8     | 44.6±11.4     | 47.0± 6.2            |
> | 10m11m | 82.1± 4.3     | 81.4± 9.7     | 80.4±3.1      | 72.3±13.8     | **86.3±14.6**        |
> | 10m12m | 11.5± 6.9     | 9.4± 5.9      | 8.1± 2.2      | 7.3± 5.5      | **14.5± 9.1**        |
> | 13m15m | 0.2± 0.3      | 0.4± 0.6      | 0.6± 0.5      | 0.4± 0.6      | **1.3± 2.5**         |
>
> ***Sensitivity Analysis on $\alpha$***
> *Source Tasks*
>
> | Tasks | $\alpha=1$ | $\alpha=5$     | $\alpha=20$ | $\alpha=10$ (ours) |
> | ------- | ------------ | ---------------- | ------------- | -------------------- |
> | 3m    | 99.2± 0.6 | **99.9± 0.0** | 99.8± 0.3  | 99.5± 0.3         |
> | 5m6m  | 70.8± 4.7 | **71.0± 2.9** | 66.5± 1.1  | 66.1± 7.0         |
> | 9m10m | 97.1± 1.8 | **97.9± 2.1** | 93.4± 1.6  | 95.5± 2.7         |
>
> *Target Tasks*
>
> | Tasks  | $\alpha=1$     | $\alpha=5$     | $\alpha=20$ | $\alpha=10$ (ours) |
> | -------- | ---------------- | ---------------- | ------------- | -------------------- |
> | 4m     | 98.4± 0.0     | 98.1± 0.5     | 99.0± 0.6  | **99.2± 1.2**     |
> | 5m     | **99.8± 0.3** | **99.8± 0.3** | 98.8± 1.8  | 99.2± 1.2         |
> | 10m    | **99.4± 0.5** | 99.4± 0.9     | 99.0± 0.3  | 98.4± 0.8         |
> | 12m    | 59.2±36.2     | 63.8±33.6     | 66.5±26.5  | **75.5±17.9**     |
> | 7m8m   | 34.5± 5.0     | 34.8±10.3     | 34.6± 2.8  | **35.3± 9.8**     |
> | 8m9m   | 46.3± 2.5     | **59.2±13.3** | 53.3± 3.6  | 47.0± 6.2         |
> | 10m11m | 83.3± 8.2     | 77.9±13.7     | 77.3± 6.8  | **86.3±14.6**     |
> | 10m12m | 10.2± 4.1     | 8.3± 8.3      | 11.9± 6.7  | **14.5± 9.1**     |
> | 13m15m | **2.5± 3.5**  | 0.2± 0.3      | 1.3± 1.8   | 1.3± 2.5          |
>
> **W2:** The writing needs to be significantly improved.
>
> **A:** Thanks for your reminder, we will further improve our writing.
>
> **W3:** It's not clear why the common skills can integrate cooperative temporal knowledge and enable an offline exploration.
>
> **A:** Firstly, cooperative temporal knowledge in this paper represents information for future prediction and value maximization from the global/team perspective. We train the learned skills to predict the future global state and maximize the team reward from the current local observations. Predicting the next global state from local observation requires the policy to understand the team-wise relationships, enabling the policy to extract more knowledge corresponding to the global information. Maximizing the team's accumulated value enables the policy to select the proper common skills that achieve the best cooperation performance, improving the policy's cooperation capability. Secondly, we utilize the IQL paradigm to train the common skills. This learning process greatly balances reward maximization and trajectory imitation from a global perspective, enabling the offline exploration of the cooperative multi-agent policy.

---

> ### Author Response · Authors · 2024-11-21
>
> **W4:** Since all tasks share the same space for the local skills $z^{1:K}_t$, why $z$ can embed task-specific knowledge?
>
> **A:** Referring to Eqs. (10) and (11), $z$ is regularized by a contrastive learning objective. This objective ensures $z$ inferred in different tasks possess different distributions (i.e., skills in the same tasks will become closer, and in different tasks will be pushed away). Therefore, $z$ represents the task-specific knowledge.
>
> **W5:** Multi-agent skills should provide abstractions at both temporal-level and subgroup-level. That is, each multi-agent skill should embed a (short) control sequence of a subgroup of agents. How is this aspect shown in the algorithm design?
>
> **A:** Although we do not explicitly divide the agents into several subgroups, their common skills learn team-wise dynamics by predicting the common next global state and improve multi-agent cooperation by maximizing the team's accumulated rewards, achieving the temporal-level and team-level cooperative policy learning. Therefore, learning the common and task-specific skills is effective for offline multi-task MARL.
>
> **W6:** Detailed instructions on how to reproduce the results in this paper using the released code.
>
> **A:** Thanks for your suggestion, we have revised the instructions. Our code in the SMAC benchmark is based on ODIS's official code. Thanks to the clear instructions in the ODIS source, we make small changes in instructions to fit our method, experiments and can be reproduced based on current instructions. We will acknowledge ODIS in our official repository. In MAMuJoCo experiments, we produce our method based on OMIGA's [1] code rather than pymarl framework, which is different from the SMAC's experiments. We will release it in our official repository.
>
> **Reference**
>
> [1] 'Offline Multi-Agent Reinforcement Learning with Implicit Global-to-Local Value Regularization'

---

> > ### Comment · Reviewer_sVE6 · 2024-12-01
> >
> > Thank you for your response. This paper builds upon ODIS, and I believe its main contribution lies in achieving enhanced performance on the current benchmarks. However, the clarity of the methodology section and the reproducibility of the results could be further improved.
> >
> > I will raise my rating to 6.

---

> > > ### Author Response · Authors · 2024-12-02
> > >
> > > Thank you very much for your kind response and support of our paper.
> > >
> > > We will further enhance the clarity of the methodology section. For example, we consider adding a brief version of the pseudo-code (Algorithm 1) into Section 3 to help explain the training and inference pipeline of our method. We also consider adding more implementation details along with a detailed illustration of our framework (Figure 4) in the appendix. If you have any other detailed suggestions regarding the clarity of the writing, please feel free to mention them. We are grateful for these suggestions to further improve the presentation of our paper.
> > >
> > > Regarding reproducibility, following your suggestions, we updated all the code and instructions to reproduce the SMAC experiments. We promise to open-source both the code and the dataset for MAMuJoCo in the future.
> > >
> > > Thank you again for taking the time to review our paper and for your valuable comments.

---

### Official Review · Reviewer_94BV · 2024-11-01

**Soundness:** 3
**Presentation:** 3
**Contribution:** 2
**Rating:** 6
**Confidence:** 4

**Summary:**

This paper introduces Hierarchical and Separate Skill Discovering (HiSSD), a method for learning generalizable skills in offline multi-task multi-agent reinforcement learning. HiSSD's hierarchical framework simultaneously learns common skills (cooperative behaviors across tasks) and task-specific skills (unique task characteristics). It uses an objective function combining reward maximization and next-state prediction for common skills, and contrastive learning for task-specific skills. HiSSD trains on offline data and enables decentralized execution with local information during testing.

**Strengths:**

This paper introduces Hierarchical and Separate Skill Discovering (HiSSD), a method for learning generalizable skills in offline multi-task multi-agent reinforcement learning. HiSSD's key strength is its hierarchical framework that simultaneously learns common and task-specific skills, overcoming existing limitations. It enables fine-grained action execution and trains using only offline data. During testing, HiSSD allows decentralized execution using local information and learned skills without centralized control. Experiments on SMAC and MAMuJoCo benchmarks showed HiSSD's performance improvements, especially in unseen tasks.

**Weaknesses:**

While the paper claims to integrate cooperative temporal knowledge into the process of learning common skills, it fails to adequately explain the clear causal relationship or mechanism between these two concepts. There appears to be a logical leap in assuming that simply performing global state prediction and value estimation directly leads to the learning of cooperative knowledge. Furthermore, the paper lacks a clear definition of what exactly "cooperative temporal knowledge" means and how it is measured and evaluated. This ambiguity makes it difficult to verify whether this knowledge has been learned.

**Questions:**

* Can you explain how performing global state prediction and value estimation directly leads to learning cooperative knowledge?
* HiSSD claims to simultaneously learn common skills and task-specific skills, but how did you verify how distinct these two types of skills actually are? Was there any overlap or interference between the two skills?

---

> ### Author Response · Authors · 2024-11-21
> **Reply to reviewer 94BV**
>
> Thank you for your time and feedback on HiSSD. We address each of your questions below.
>
> ### Weaknesses:
>
> **W1:** This paper didn't adequately explain the clear causal relationship or mechanism between integrating cooperative temporal knowledge and learning common skills.
>
> **A:** Theoretically, we treat the offline policy learning as probabilistic inference, formulating the trajectory distributions induced by the optimal policy and our trained policy as $p(\tau)$ in Eq. (2) and $\hat{p}(\tau)$ in Eq. (3), respectively. Our goal is to minimize the distance between these two distributions and the formal objective is given by Eq. (4), where the multi-agent policy needs to maximize the accumulated rewards and imitate the trajectory sampled from the offline dataset based on selected common skills. The first term in Eq. (4) is to learn common skills that maximize team rewards. This term benefits the multi-agent policy to achieve cooperation by selecting proper common skills. The second term in Eq. (4) is to predict the next global state based on common skills. This term helps to integrate both temporal-wise and team-wise knowledge into common skills. From human intuition, a superior cooperation multi-agent policy must achieve the same goal and understand the dynamic transitions from the team's perspective. Therefore we designed the proposed learning objective.
>
> ---
> ### Questions:
>
> **W2 & Q1:** Explain how performing global state prediction and value estimation directly leads to learning cooperative knowledge.
>
> **A:** In this paper, cooperative temporal knowledge represents information for future state prediction and value maximization from the global/team perspective. In RL, temporal knowledge is highly related to the transition $(s_t, a_t, r_t, s_{t+1})$, which has been widely discussed in prior works [1, 2]. In MARL, predicting the next global state from local observation requires the policy to understand the team-wise relationships, enabling the policy to extract more knowledge corresponding to the global information. Maximizing the team's accumulated value enables the policy to select the proper common skills that achieve the best cooperation performance, improving the policy's cooperation capability.
>
> **Q2:** How did the authors verify how distinct these two types of skills are? Was there any overlap or interference between the two skills?
>
> **A:** The definition and learning process of these two skills are significantly different. The common skill is concatenated with the current observation to infer the observation embedding, while the specific skill is fed to the action output layer to guide the action execution. To clearly show the distinction between these two types of skills we visualize their distribution using t-SNE and present them in Fig. (5) in the revised paper. The empirical results indicate that the learned policy can distinguish different kinds of skills at each stage of the episode, and no obvious overlap between these skills is observed.
>
> **Reference**
>
> [1] 'Bridging State and History Representations: Understanding Self-Predictive RL'
>
> [2] 'Joint-Predictive Representations for Multi-Agent Reinforcement Learning'

---

> > ### Comment · Reviewer_94BV · 2024-11-25
> > **Official Comment by Reviewer 94BV**
> >
> > Thank you to the authors for addressing my questions. It seems that my concerns have been resolved. Overall, I find the proposed framework to be promising, and I have decided to maintain my original score.

---

> > > ### Author Response · Authors · 2024-11-26
> > >
> > > Thank you for taking the time to review our paper and for your support!

---

### Official Review · Reviewer_7LGd · 2024-11-03

**Soundness:** 4
**Presentation:** 4
**Contribution:** 3
**Rating:** 6
**Confidence:** 3

**Summary:**

This paper proposes a hierarchical framework, HiSSD, for jointly learning common and task-specific skills from offline multi-task datasets. The common skills represent the general cooperation patterns, while task-specific skills achieve a task-guided fine-grained action execution. The algorithm is broken up into two phases. In the first phase, the high-level planner discovers common skills. In the second stage, the low-level controller learns task-specific skills.

**Strengths:**

1. The paper is well written, with thorough experiments, and a variety of tasks across two environments.
2. In order to achieve better performances in cooperative multi-agent environments, the paper brings cooperative temporal knowledge into common skills, which is reasonable.

**Weaknesses:**

1. According to the paper, HiSSD is superior to previous skill-based methods like ODIS because it brings cooperative temporal knowledge into common skills and discovers task-specific skills that can distinguish each task’s specific knowledge. However, HiSSD fails to outperform ODIS in some tasks.
2. In MAMuJoCo, the paper didn't use ODIS as a baseline, which has a good performance in SMAC.
3. The process of learning skills is similar to imitation learning, so it failed to perform well in unseen tasks, indicating low-level generalization ability. For example, in Table 1, the win rates for 10m12m and 13m15m are lower than 10% with the medium-quality dataset.
4. In the passage, some verbs sound a bit odd. For example, 'contain' in the sentence 'Equipping offline multi-task MARL with skill learning to improve policy transfer still contains issues.'
5. One concern for the paper is to evaluate HiSSD in maps that are more difficult (like the Stalker-Zealot task set in ODIS), to validate its effectiveness.

**Questions:**

- In Table 3, I wonder why presents BC-best's results, rather than ODIS. ODIS seems to be the baseline with the best performances.
- In Figure 3, Large-scale tasks (i.e., 10m and 12m) overlap with each other but small-scale tasks don't. The paper said this is because 10m and 12m are similar. However, 3m, 4m, and 5m are also similar. I would appreciate some further discussion about this.
- In Equation 6, the weight $\alpha$ serves as the trade-off between guiding to space with high-reward and space that the execution policy ought to have a correct imitation. I wonder why the paper didn't conduct a sensitive analysis for this parameter like $\beta$ in Equation 11, as shown in Appendix E.2.
- The task-specific skills learn task-specific knowledge of source tasks, why do these skills play an important role in unseen tasks?

---

> ### Author Response · Authors · 2024-11-21
> **Reply to reviewer 7LGd**
>
> Thank you for your time and feedback on HiSSD. We address each of your questions below.
>
> ### Weaknesses:
>
> **W1:** Although HiSSD is superior to previous skill-based methods, it fails to outperform ODIS in some tasks.
>
> **A:** For the performance in marine task sets in SMAC, HiSSD outperforms ODIS on most tasks. HiSSD obtains the best scores in 68 tasks while ODIS obtains the best in 17. HiSSD fails to outperform ODIS mainly in a few large-scale tasks (e.g., 10m12m and 13m15m). This is due to 1) the distribution distance between large-scale tasks and source training tasks is farther than other target tasks. It is much more difficult for the policy to generalize skills to the large-scale tasks. 2) These large-scale tasks are more complex and the number of units is more asymmetric. All of the current methods are unable to obtain convincing performance (average win rates are under 15%).
>
> | Total Tasks | Others | ODIS | HiSSD |
> | ------------- | -------- | ------ | ------- |
> | 88          | 3      | 17   | 68    |
>
> **W2:** In MAMuJoCo, the paper didn't use ODIS as a baseline.
>
> **A:** It's worth noting that the official code released by ODIS didn't conduct experiments on MAMuJoCo. Therefore, we didn't use ODIS as a baseline. Here we reproduce ODIS on MAMuJoCo environment and present the results below. For a fair comparison, we use the same architecture and hyperparameters as HiSSD. The empirical results indicate that HiSSD is competitive in source tasks and outperforms in target tasks compared to ODIS, which demonstrates the better generalization of HiSSD.
>
> ***Reproduced ODIS on MAMuJoCo***
> *Source Tasks*
>
> | Tasks       | BC              | ODIS            | HiSSD               |
> | ------------- | ----------------- | ----------------- | --------------------- |
> | complete    | 3188.16±566.68 | 3677.66±174.82 | **4450.57±126.36** |
> | back thigh  | 3324.22± 58.49 | 2381.62±198.59 | **3698.38± 13.98** |
> | back foot   | 3079.01±355.66 | 2713.40±195.63 | **3197.83± 6.99**  |
> | front thigh | 1861.53±415.80 | **2684.99±249.70** | 1948.74± 81.24 |
> | front shin  | 1819.94±273.96 | **3944.78±219.72** | 3468.32±290.72 |
>
> *Target Tasks*
>
> | Tasks      | BC              | ODIS                | HiSSD               |
> | ------------ | ----------------- | --------------------- | --------------------- |
> | back shin  | 1964.55±268.24 | 3217.59±184.15 | **3472.12± 91.95**     |
> | front foot | 3468.40±369.40 | 3930.82±342.16     | **4165.29±338.96** |
>
> **W3:** The process of learning skills is similar to imitation learning, so it failed to perform well in unseen tasks (e.g., 10m12m and 13m15m), indicating low-level generalization ability.
>
> **A:** Generalizing the multi-task offline policy to the unseen large-scale tasks (e.g., 10m12m and 13m15m) is more difficult: 1) the number of units in these tasks are more asymmetric than other tasks and 2) the task distributions are farther from source tasks than other unseen tasks. Therefore, it is difficult for the policy to extract useful skills and behavior patterns from the proposed offline multi-task dataset, leading to low absolute win rates. We consider improving policy's generalization capability in large-scale tasks as our future work.
>
> **W4:** In the passage, some verbs sound a bit odd.
>
> **A:** Thanks for your reminder, we will check and improve our writing.

---

> ### Author Response · Authors · 2024-11-21
>
> **W5:** One concern for the paper is to evaluate HiSSD in maps that are more difficult (e.g., Stalker-Zealot task set in ODIS), to validate its effectiveness.
>
> A: To further validate HiSSD's effectiveness, we conduct experiments on 'Stalker-Zealot' task set proposed by ODIS and present the averaged results below. Detailed results are presented in Appendix (F.2) in the revised paper. It should be noted that the stalker-zealot dataset proposed by ODIS is not well developed, which may hinder the training and generalization of the policy. Based on the results, we find no method is consistently best. HiSSD outperforms all baselines in half of the target tasks while still being competitive in source tasks, indicating the better generalization capability of hiSSD.
>
> ***Average test win rates in the Stalker-Zealot task set with different data qualities.***
> *Source Tasks*
>
> | Data Qualities | BC-best    | UPDeT-m        | ODIS           | HiSSD          |
> | ---------------- | ------------ | ---------------- | ---------------- | ---------------- |
> | Expert         | 87.9± 5.6 | 30.2±26.6     | 82.0± 6.3     | **89.3± 4.0** |
> | Medium         | 28.6±10.1 | 26.5±19.2     | **37.0± 9.1** | 24.6± 7.3     |
> | Medium-Expert  | 52.9±17.9 | 48.5±25.1     | 47.1±13.9     | **55.9± 9.7** |
> | Medium-Replay  | 13.2± 5.4 | **15.6±13.2** | 14.1± 8.8     | 10.8± 1.6     |
>
> *Target Tasks*
>
> | Data Qualities | BC-best     | UPDeT-m     | ODIS           | HiSSD          |
> | ---------------- | ------------- | ------------- | ---------------- | ---------------- |
> | Expert         | 60.9±19.0  | 13.8±13.2  | 62.8± 7.5     | **69.2± 8.7** |
> | Medium         | 19.8± 14.1 | 10.1± 11.8 | **24.7± 8.4** | 16.6± 7.4     |
> | Medium-Expert  | 37.5±17.4  | 18.6±14.1  | 38.2±11.9     | **40.3±12.8** |
> | Medium-Replay  | 13.6±7.6   | 10.7±11.8  | 11.0±10.8     | **13.6± 4.8** |
>
> ---
> ### Questions:
>
> **Q1:** Why present BC-best's results in Table 3.
>
> **A:** Because our method is built on the BC-based method, we use BC-best rather than ODIS as a baseline in the ablation study to show the effectiveness of different components in HiSSD.
>
> **Q2:** In Fig. (3), Large-scale tasks (i.e., 10m and 12m) overlap with each other but small-scale tasks don't. The paper said this is because 10m and 12m are similar. However, 3m, 4m, and 5m are also similar.
>
> **A:** Our training objective ensures that the policy is able to distinguish task-specific skills when the target tasks are close to the distribution of source tasks. For the small-scale tasks that are similar to the source tasks, the policy can be easily generalized to the small-scale target tasks, and the discrepancy between these tasks can be learned more efficiently. As for the large-scale tasks (i.e., 10m and 12m) which are farther from the distribution of the source tasks, it is harder for the policy to capture significant discrepancies and it can only infer some knowledge related to the source tasks, therefore causing the clearer overlap.
>
> **Q3:** Why the paper didn't conduct a senstive analysis for $\alpha$ in Eq. (6).
>
> **A:** We set $\alpha$ and other hyperparameters following previous works. To better analyze the model sensitivity of our method, we conduct an additional ablation study under different $\alpha$ and present the results below. The results indicate that our method is robust to $\alpha$.
>
> ***Sensitivity Analysis on $\alpha$ in Marine-hard Task Set***
> *Source Tasks*
>
> | Tasks | $\alpha=1$ | $\alpha=5$     | $\alpha=20$ | $\alpha=10$ (ours) |
> | ------- | ------------ | ---------------- | ------------- | -------------------- |
> | 3m    | 99.2± 0.6 | **99.9± 0.0** | 99.8± 0.3  | 99.5± 0.3         |
> | 5m6m  | 70.8± 4.7 | **71.0± 2.9** | 66.5± 1.1  | 66.1± 7.0         |
> | 9m10m | 97.1± 1.8 | **97.9± 2.1** | 93.4± 1.6  | 95.5± 2.7         |
>
> *Target Tasks*
>
> | Tasks  | $\alpha=1$     | $\alpha=5$     | $\alpha=20$ | $\alpha=10$ (ours) |
> | -------- | ---------------- | ---------------- | ------------- | -------------------- |
> | 4m     | 98.4± 0.0     | 98.1± 0.5     | 99.0± 0.6  | **99.2± 1.2**     |
> | 5m     | **99.8± 0.3** | **99.8± 0.3** | 98.8± 1.8  | 99.2± 1.2         |
> | 10m    | **99.4± 0.5** | 99.4± 0.9     | 99.0± 0.3  | 98.4± 0.8         |
> | 12m    | 59.2±36.2     | 63.8±33.6     | 66.5±26.5  | **75.5±17.9**     |
> | 7m8m   | 34.5± 5.0     | 34.8±10.3     | 34.6± 2.8  | **35.3± 9.8**     |
> | 8m9m   | 46.3± 2.5     | **59.2±13.3** | 53.3± 3.6  | 47.0± 6.2         |
> | 10m11m | 83.3± 8.2     | 77.9±13.7     | 77.3± 6.8  | **86.3±14.6**     |
> | 10m12m | 10.2± 4.1     | 8.3± 8.3      | 11.9± 6.7  | **14.5± 9.1**     |
> | 13m15m | **2.5± 3.5**  | 0.2± 0.3      | 1.3± 1.8   | 1.3± 2.5          |

---

> ### Author Response · Authors · 2024-11-21
>
> **Q4:** The task-specific skills learn task-specific knowledge of source tasks, why do these skills play an important role in unseen tasks?
>
> **A:** Common skills represent the general knowledge that can be reused in different tasks. Meanwhile learning task-specific skills aims to discover knowledge related to the current task, achieving complementarity with common skills. Moreover, we conduct a variant of HiSSD named 'w/o Specific' which only learns common skills. We evaluate it on Marine-hard task set and present the results below. The empirical results indicate that the task-specific skills further improve the policy's generalization capability.
>
> ***Components Analysis on Task-Specific Skills:*** 'w/o Specific' indicates only integrating common skills in HiSSD.
> *Source Tasks*
>
> | Data Qualities | w/o Specific   | BC-best    | HiSSD          |
> | ---------------- | ---------------- | ------------ | ---------------- |
> | Expert         | **85.9±16.4** | 81.1±21.8 | 84.5±17.1     |
> | Medium         | 53.1±20.0     | 50.3±20.1 | **55.9±19.2** |
> | Medium-Expert  | 63.5±26.1     | 41.7±18.5 | **64.2±23.5** |
> | Medium-Replay  | 47.7±25.1     | 46.5±26.7 | **51.9±24.0** |
>
> *Target Tasks*
>
> | Data Qualities | w/o Specific | BC-best    | HiSSD          |
> | ---------------- | -------------- | ------------ | ---------------- |
> | Expert         | 55.0±37.4   | 46.8±36.8 | **61.2±37.4** |
> | Medium         | 37.6±34.2   | 25.6±26.8 | **48.6±40.1** |
> | Medium-Expert  | 51.2±34.3   | 38.8±33.0 | **57.6±37.5** |
> | Medium-Replay  | 42.8±36.2   | 41.2±33.7 | **48.7±39.0** |

---

### Official Review · Reviewer_niW9 · 2024-11-03

**Soundness:** 3
**Presentation:** 2
**Contribution:** 2
**Rating:** 6
**Confidence:** 4

**Summary:**

This paper proposes HiSSD to leverage a hierarchical framework to jointly learn the common and task-specific skills in offline MARL settings. Experiments in SMAC and multi-agent MuJoCo show its effectiveness and generalizability to different tasks.

**Strengths:**

1.	The division into common and task-specific skills provides a balance between generalization and task adaptation, addressing the limitations of previous methods that only focused on common skills.
2.	The proposed objective to learn common skills for exploration and prediction is new.
3.	HiSSD enables multi-agent policy transfer across varying tasks, outperforming existing multi-task MARL baselines on two benchmarks

**Weaknesses:**

1.	HiSSD’s hierarchical design and multi-component training can lead to instability. I think the authors should reveal more about the training details, hyperparameters selection, and tuning methods.
2.	It can be better to present an algorithm of the whole training process and the execution process for understanding.
3.	In the ablation studies, the HiSSD variants (e.g., w/o Planning and Half-Negative) often show strong performance compared to HiSSD with minor differences or even surpassing HiSSD.
4.	The visualization experiments do not show clear evidence about the latent skills. For example, the task-specific skills in the same task seem closer in Figure 3.

**Questions:**

1.	From Equation (3), does $q(s_{t+1}\mid c_t^{1:K})$ has any meaning in real? How does HiSSD model the sequence $c_t^{1:K}$?
2.	If the negative samples are drawn from limited seen tasks when doing task-specific skill learning, how can we ensure a correct skill when facing unseen task distribution?
3.	Why are the two baselines, ITD3-CQL and ITD3-BC, not compared in the SMAC experiments?
4.	How is the performance if we only adopt a common skill without using the task-specific skills?

---

> ### Author Response · Authors · 2024-11-21
> **Reply to reviewer niW9**
>
> Thank you for your time and feedback on HiSSD. We address each of your questions below.
>
> ### Weaknesses:
>
> **W1:** HiSSD should reveal more about training details as the framework design may lead to instability.
>
> **A:** For a fair comparison, most of the hyperparameters in our method are consistent with previous methods. To further analyze the sensitivity of our method, we add additional ablation studies and present the results below. We also added more training details in Appendix (D) in the revision. The empirical results indicate that our method is robust under different hyperparameter settings. This makes it easier for our method to tune hyperparameters. We keep the same settings as previous methods for a simple reproduction.
>
> ***Sensitivity Analysis on $\epsilon$ in Marine-hard Task Set***
> *Source Tasks*
>
> | Tasks | $\epsilon=0.1$ | $\epsilon=0.3$  | $\epsilon=0.5$ | $\epsilon=0.7$ | $\epsilon=0.9$ (ours) |
> | ------- | ---------------- | ----------------- | ---------------- | ---------------- | ----------------------- |
> | 3m    | 99.6 ± 0.6    | **99.9 ± 0.0** | 99.4 ± 0.5    | 99.4 ± 0.5    | 99.5 ± 0.3           |
> | 5m6m  | 73.5± 5.1     | 71.3± 3.3      | 70.8± 2.4     | **74.4± 1.5** | 66.1± 7.0            |
> | 9m10m | 98.3± 0.3     | 97.1± 0.3      | **99.0± 0.3** | 96.7± 0.6     | 95.5± 2.7            |
>
> *Target Tasks*
>
> | Tasks  | $\epsilon=0.1$ | $\epsilon=0.3$ | $\epsilon=0.5$ | $\epsilon=0.7$ | $\epsilon=0.9$ (ours) |
> | -------- | ---------------- | ---------------- | ---------------- | ---------------- | ----------------------- |
> | 4m     | 97.7± 1.6     | 99.0± 0.3     | 99.0± 0.8     | 98.4± 0.5     | **99.2± 1.2**        |
> | 5m     | **99.8± 0.3** | **99.8± 0.3** | 99.2± 0.3     | 99.1± 1.2     | 99.2± 1.2            |
> | 10m    | 99.2± 0.8     | 98.5± 1.6     | 99.2± 0.3     | **99.2± 0.3** | 98.4± 0.8            |
> | 12m    | 69.0±33.7     | 67.9±17.0     | 71.0±26.0     | **87.5± 7.6** | 75.5±17.9            |
> | 7m8m   | 29.2± 2.8     | 23.1± 4.9     | 32.7± 3.8     | 25.4± 2.9     | **35.3± 9.8**        |
> | 8m9m   | 46.0±16.6     | **47.5± 9.2** | 45.4±14.8     | 44.6±11.4     | 47.0± 6.2            |
> | 10m11m | 82.1± 4.3     | 81.4± 9.7     | 80.4±3.1      | 72.3±13.8     | **86.3±14.6**        |
> | 10m12m | 11.5± 6.9     | 9.4± 5.9      | 8.1± 2.2      | 7.3± 5.5      | **14.5± 9.1**        |
> | 13m15m | 0.2± 0.3      | 0.4± 0.6      | 0.6± 0.5      | 0.4± 0.6      | **1.3± 2.5**         |
>
> ***Sensitivity Analysis on $\alpha$ in Marine-hard Task Set***
> *Source Tasks*
>
> | Tasks | $\alpha=1$ | $\alpha=5$     | $\alpha=20$ | $\alpha=10$ (ours) |
> | ------- | ------------ | ---------------- | ------------- | -------------------- |
> | 3m    | 99.2± 0.6 | **99.9± 0.0** | 99.8± 0.3  | 99.5± 0.3         |
> | 5m6m  | 70.8± 4.7 | **71.0± 2.9** | 66.5± 1.1  | 66.1± 7.0         |
> | 9m10m | 97.1± 1.8 | **97.9± 2.1** | 93.4± 1.6  | 95.5± 2.7         |
>
> *Target Tasks*
>
> | Tasks  | $\alpha=1$     | $\alpha=5$     | $\alpha=20$ | $\alpha=10$ (ours) |
> | -------- | ---------------- | ---------------- | ------------- | -------------------- |
> | 4m     | 98.4± 0.0     | 98.1± 0.5     | 99.0± 0.6  | **99.2± 1.2**     |
> | 5m     | **99.8± 0.3** | **99.8± 0.3** | 98.8± 1.8  | 99.2± 1.2         |
> | 10m    | **99.4± 0.5** | 99.4± 0.9     | 99.0± 0.3  | 98.4± 0.8         |
> | 12m    | 59.2±36.2     | 63.8±33.6     | 66.5±26.5  | **75.5±17.9**     |
> | 7m8m   | 34.5± 5.0     | 34.8±10.3     | 34.6± 2.8  | **35.3± 9.8**     |
> | 8m9m   | 46.3± 2.5     | **59.2±13.3** | 53.3± 3.6  | 47.0± 6.2         |
> | 10m11m | 83.3± 8.2     | 77.9±13.7     | 77.3± 6.8  | **86.3±14.6**     |
> | 10m12m | 10.2± 4.1     | 8.3± 8.3      | 11.9± 6.7  | **14.5± 9.1**     |
> | 13m15m | **2.5± 3.5**  | 0.2± 0.3      | 1.3± 1.8   | 1.3± 2.5          |
>
> **W2:** It can be better to present an algorithm of the whole training process and the execution process for understanding.
>
> **A:** Thanks for this suggestion. We have presented the pseudocode in Appendix (B), which shows the brief training and execution process of our method.
>
> **W3:** In the ablation studies, the HiSSD variants (e.g., w/o Planning and Half-Negative) often show strong performance compared to HiSSD with minor differences or even surpassing HiSSD.
>
> **A:** Firstly, the Half-Negative variant in the ablation studies shares the same architecture and objectives with the original HiSSD while reducing the number of negative samples. Comparing HiSSD with this variant demonstrates that more negative samples will lead to better performance. Secondly, although the original HiSSD shows minor differences in some source tasks compared to other variants, it significantly performs best in target tasks, indicating the improvement of generalization capability.

---

> ### Author Response · Authors · 2024-11-21
>
> **W4:** The visualization experiments do not show clear evidence about the latent skills. For example, the task-specific skills in the same task seem closer in Fig. (3).
>
> **A:** We visualize the learned common and task-specific skills in Figs. (2) and (3) respectively. Fig. (2) indicates that the learned common skills will be clustered based on cooperative patterns rather than tasks. Fig. (3) shows the distribution of the task-specific skills in different time steps. Task-specific skills are trained to represent the task-relevant knowledge, serving as the common skills' complementary and enhancing the adaptive action execution. Therefore, task-specific skills in the same tasks should be clustered together. Moreover, as steps go on, different tasks in the same task set will degrade to various sub-tasks that are more similar. The distance between specific skills' distribution from the middle & end stages of each episode will be closer compared to the distribution in the early stage.
>
> ---
>
> ### Questions:
>
> **Q1:** From Eq. (3), does $q(s_{t+1} \vert c_t^{1:K})$ has any meaning in real? How does HiSSD model the sequence $c_t^{1:K}$?
>
> **A:** In Eq. (3), $s_{t+1}$ represent the ground-truth global next state and $q(s^\prime_{t+1} \vert c_t^{1:K})$ indicates that we use a forward predictor $q$ to predict the next global state using all agents' common skills. For example, in the SMAC environment, $q(s^\prime_{t+1} \vert \cdot)$ indicates using common skills selected by each unit to estimate the whole battlefield in the next time step. This process can be formulated as $s^\prime_{t+1} = q(c_t^{1:K})$, where $c_t^{1:K} = \pi_h (o_t^{1:K})$ is each agent's common skill. These skills are inferred by a transformer-based policy $\pi_h$ based on their local observation $o_t^{1:K}$ and ${1:K}$ indicates agents' index. The shape of $c_t^{1:K}$ in SMAC and MAMuJoCo environments are $(n_\mathrm{agent}, n_\mathrm{entity},  n_\mathrm{hidden})$ and $(n_\mathrm{agent}, n_\mathrm{hidden})$, respectively.
>
> **Q2:** How can we ensure a correct skill under unseen task distribution as the negative samples are drawn from the limited seen task.
>
> **A:** We train the task-specific skills to extract potential complementary knowledge for common skills and identify various behavior patterns for potential sub-tasks from the multi-task offline data. Although the target tasks are unknown, the policy is able to extract certain behavior patterns learned from multi-tasks that are more related to the current task. Our prime goal is to demonstrate that compared to the methods in literature, integrating task-specific skills into offline multi-task learning can further improve the policy's generalization capability. Moreover, we conduct a variant of HiSSD without task-specific skills (named w/o specific) additional experiments as the ablation study and represent the results below in answer to **Q4**. The results indicate the effectiveness of task-specific skills when deploying the skill-based policy in unseen tasks.
>
> **Q3:** Why are the two baselines, ITD3-CQL and ITD3-BC, not compared in the SMAC experiments?
>
> **A:** It's worth noting that ITD3-CQL and ITD3-BC are built on TD3 algorithm, which is typically designed for continuous control tasks. Therefore, these baselines can not be directly applied to the SMAC environment with discrete action space.
>
> **Q4:** How is the performance if we only adopt a common skill without using the task-specific skills?
>
> **A:** We conduct a variant of HiSSD named 'w/o specific' which does not learn task-specific skills and report the experimental results below. We find that this variant obtained significant improvement compared to the behavior cloning baseline (BC), indicating the benefits of our common skills learning pipeline. Moreover, the original HiSSD outperforms the w/o Specific variant, demonstrating that learning task-specific skills from a multi-task dataset can further improve the policy's generalization capability.
>
> ***Components Analysis on Task-Specific Skills:*** 'w/o Specific' indicates only integrating common skills in HiSSD.
> *Source Tasks*
>
> | Data Qualities | w/o Specific   | BC-best    | HiSSD          |
> | ---------------- | ---------------- | ------------ | ---------------- |
> | Expert         | **85.9±16.4** | 81.1±21.8 | 84.5±17.1     |
> | Medium         | 53.1±20.0     | 50.3±20.1 | **55.9±19.2** |
> | Medium-Expert  | 63.5±26.1     | 41.7±18.5 | **64.2±23.5** |
> | Medium-Replay  | 47.7±25.1     | 46.5±26.7 | **51.9±24.0** |
>
> *Target Tasks*
>
> | Data Qualities | w/o Specific | BC-best    | HiSSD          |
> | ---------------- | -------------- | ------------ | ---------------- |
> | Expert         | 55.0±37.4   | 46.8±36.8 | **61.2±37.4** |
> | Medium         | 37.6±34.2   | 25.6±26.8 | **48.6±40.1** |
> | Medium-Expert  | 51.2±34.3   | 38.8±33.0 | **57.6±37.5** |
> | Medium-Replay  | 42.8±36.2   | 41.2±33.7 | **48.7±39.0** |

---

> > ### Comment · Reviewer_niW9 · 2024-11-26
> >
> > I am generally satisfied with the authors’ rebuttal and have decided to raise my score to 6.

---

> > > ### Author Response · Authors · 2024-11-26
> > >
> > > Thank you for your valuable comments and suggestions!

---

### Official Review · Reviewer_yFSo · 2024-11-04

**Soundness:** 3
**Presentation:** 3
**Contribution:** 3
**Rating:** 6
**Confidence:** 4

**Summary:**

This paper proposes a new method called Hierarchical and Separate Skill Discovering (HiSSD) for learning generalizable multi-agent policies from offline multi-task data. The key idea is to jointly learn common skills that capture cooperative behaviors across tasks, as well as task-specific skills for fine-grained action execution. The method uses a hierarchical framework with a high-level planner that learns common skills and a low-level controller that learns task-specific skills. Experiments on SMAC and Multi-Agent MuJoCo benchmarks show improved generalization to unseen tasks compared to baselines. Overall, this paper presents a novel and effective approach to an important problem in multi-agent RL. With some additional analysis and clarifications, it would make a good contribution to the field.

**Strengths:**

1. Novel approach to learning generalizable multi-agent policies by jointly learning common and task-specific skills.

2. Theoretical analysis and derivation of objectives for skill learning.

3. Comprehensive experiments on multiple benchmarks showing improved generalization.

4. Addresses an important problem in multi-agent RL of transferring to tasks with varying numbers of agents.

5. The authors have shared comprehensive details of implementation along with codebase, which further bolsters the experimental efforts.

**Weaknesses:**

1. Ablation study can be further expanded to have a comprehensive analysis of different components. Comparison to some recent related works like HyGen is missing.

**Questions:**

1. There is a very recent work that tackles the same problem from a different methodology perspective found here : "Variational Offline Multi-agent Skill Discovery" by Chen et. al. I would like to know the authors' thoughts on how their methodology is different from this work, and if there are scenarios where one or the other might work well/worse.

---

> ### Author Response · Authors · 2024-11-21
> **Reply to reviewer yFSo**
>
> Thank you for your time and feedback on HiSSD. We address each of your questions below.
>
> ### Weaknesses:
>
> **W1:** Ablation study can be further expanded to have a comprehensive analysis of different components.
>
> **A:** We have already conducted several ablation studies to analyze the effectiveness of different components in our paper. To further show the benefits of task-specific skills, we conduct a variant of HiSSD named 'w/o Specific' which only learns common skills. We evaluate it on SMAC's Marine-hard task set and present the results below. The empirical results indicate the effectiveness of task-specific skills in offline multi-task MARL.
>
> ***Components Analysis on Task-Specific Skills:*** 'w/o Specific' indicates only integrating common skills in HiSSD.
> *Source Tasks*
>
> | Data Qualities | w/o Specific   | BC-best    | HiSSD          |
> | ---------------- | ---------------- | ------------ | ---------------- |
> | Expert         | **85.9±16.4** | 81.1±21.8 | 84.5±17.1     |
> | Medium         | 53.1±20.0     | 50.3±20.1 | **55.9±19.2** |
> | Medium-Expert  | 63.5±26.1     | 41.7±18.5 | **64.2±23.5** |
> | Medium-Replay  | 47.7±25.1     | 46.5±26.7 | **51.9±24.0** |
>
> *Target Tasks*
>
> | Data Qualities | w/o Specific | BC-best    | HiSSD          |
> | ---------------- | -------------- | ------------ | ---------------- |
> | Expert         | 55.0±37.4   | 46.8±36.8 | **61.2±37.4** |
> | Medium         | 37.6±34.2   | 25.6±26.8 | **48.6±40.1** |
> | Medium-Expert  | 51.2±34.3   | 38.8±33.0 | **57.6±37.5** |
> | Medium-Replay  | 42.8±36.2   | 41.2±33.7 | **48.7±39.0** |
>
> **W2:** Comparison to some recent related works like HyGen is missing.
>
> **A:** It's worth noting that the problem setting of HyGen is different from HiSSD. HyGen attempts to integrate offline pre-training for skill discovery to enhance the **online multi-task learning**, while HiSSD aims to improve the policy's generalization capability by **offline multi-task learning**. HyGen first pre-trains the skill space with an action decoder using the global information, then conducts additional **online exploration** to maintain a hybrid replay buffer using offline and online data, improving the high-level policy and refining the action decoder simultaneously. This helps HyGen to correct the mistakes in value estimation when learning from poor offline data. Compared to HyGen, our method requires **no online exploration** and pretraining steps. HyGen only conducts common skills learning while our method leverages task-specific skills to complement the common skills. We compare our method to HyGen in the SMAC's Marine hard task set and present the experimental results below. When the offline data quality is good (i.e., Expert), HiSSD outperforms HyGen on most tasks, showing its good generalization of learning from offline multi-task data. When the data quality is medium, HyGen benefits from additional online learning and further improves the policy.
>
> ***Expert data quality***
> *Source tasks*
>
> | Tasks | BC-best    | HyGen          | HiSSD          |
> | ------- | ------------ | ---------------- | ---------------- |
> | 3m    | 97.7± 2.6 | 99.1± 1.0     | **99.5± 0.3** |
> | 5m6m  | 50.4± 2.3 | 61.2± 8.0     | **66.1± 7.0** |
> | 9m10m | 95.3± 1.6 | **96.4± 3.0** | 95.5± 2.7     |
>
> *Target tasks*
>
> | Tasks  | BC-best    | HyGen          | HiSSD          |
> | -------- | ------------ | ---------------- | ---------------- |
> | 4m     | 92.1± 3.5 | 95.8± 4.0     | **99.2± 1.2** |
> | 5m     | 87.1±10.5 | **99.5± 1.0** | 99.2± 1.2     |
> | 10m    | 90.5± 3.8 | 93.5± 5.0     | **98.4± 0.8** |
> | 12m    | 70.8±15.2 | **85.2± 6.0** | 75.5±19.7     |
> | 7m8m   | 18.8± 3.1 | 28.9±12.0     | **35.3± 9.8** |
> | 8m9m   | 15.8± 3.3 | 25.7± 9.0     | **47.0± 6.2** |
> | 10m11m | 45.3±11.1 | 57.2±13.0     | **86.3±14.6** |
> | 10m12m | 1.0± 1.5  | 13.8± 4.0     | **14.5± 9.1** |
> | 13m15m | 0.0± 0.0  | **9.5± 5.0**  | 1.3± 2.5      |
>
> ***Medium data quality***
> *Source tasks*
>
> | Tasks | BC-best    | HyGen          | HiSSD      |
> | ------- | ------------ | ---------------- | ------------ |
> | 3m    | 65.4±14.7 | **91.5±11.0** | 62.7± 5.7 |
> | 5m6m  | 21.9± 3.4 | **31.6± 7.0** | 26.4± 3.8 |
> | 9m10m | 63.8±10.9 | **79.2± 4.0** | 73.9± 2.3 |
>
> *Target tasks*
>
> | Tasks  | BC-best    | HyGen          | HiSSD          |
> | -------- | ------------ | ---------------- | ---------------- |
> | 4m     | 48.8±21.1 | **91.4± 8.0** | 77.3±10.2     |
> | 5m     | 76.6±14.1 | **96.5± 6.0** | 88.4± 8.4     |
> | 10m    | 56.2±20.6 | 96.4± 3.0     | **98.0± 0.3** |
> | 12m    | 24.0±10.5 | 81.5±14.0     | **86.4± 6.0** |
> | 7m8m   | 1.6± 1.6  | 24.5± 9.0     | **14.2±10.1** |
> | 8m9m   | 3.1± 3.8  | **24.5± 9.0** | 15.3± 2.8     |
> | 10m11m | 19.7± 8.9 | **47.2±13.0** | 43.6± 4.6     |
> | 10m12m | 0.0± 0.0  | **5.2± 2.0**  | 0.6± 0.5      |
> | 13m15m | 0.6± 1.3  | **9.3± 6.0**  | 1.4± 2.4      |

---

> ### Author Response · Authors · 2024-11-21
>
> ### Questions:
>
> **Q1:** The difference between HiSSD and a very recent work 'Variational Offline Multi-agent Skill Discovery'.
>
> **A:** VO-MASD proposes an efficient method to discover cooperative patterns between agents from a subgroup view. They utilize offline datasets to jointly learn individual and subgroup skills to enhance the online MARL. Compared to VO-MASD, our method aims to train a generalizable multi-agent policy in a fully offline learning pipeline to reduce interaction costs. Our method does not learn hierarchy-like common skills, we utilize task-specific skills to complement the common skills to enhance the policy's generalization capability. Our method embeds the common skills for a weighted next-state prediction, enabling an offline policy improvement. We have added the citations of HyGen and VO-MASD in the revision.

---

### Author Response · Authors · 2024-11-25

Dear Reviewers,

Thank you for your time and effort to review our paper and provide detailed feedback. As the discussion period is coming to an end, we would like to know if our response and revision have addressed all your concerns. If so, we kindly ask for your reconsideration of the score. If there remain any concerns, we are looking forward to further discussions to address them.

We greatly appreciate your time and effort in helping improve our paper.

Best regards,

Authors

---

### Author Response · Authors · 2024-12-04

Dear Reviewers,

We sincerely thank you for taking the time to review our paper and for providing thoughtful and constructive feedback. In response to your comments, we have made several key revisions and responses below,

**Experiments**

1. We conducted additional ablation studies to analyze model sensitivity (i.e., $\alpha$ in Eq. (7) and $\epsilon$ in Eq. (5)), and to demonstrate the effectiveness of task-specific skills by comparing 'w/o Specific' with HiSSD.
2. We presented additional visualization to show the differences between our learned common and task-specific skills.
3. We performed experiments on a more challenging task set ‘Stalker Zealot’ within the SMAC environment. The results show that our method achieves the best average test win rates.
4. We reproduce the strong baseline ODIS in our MAMuJoCo task set, and the empirical results indicate that our approach outperforms ODIS.
5. We compared our method to recent work integrating online learning into multi-task learning. The results demonstrate that our method can effectively utilize offline data to learn generalizable skills across tasks

**Revisions, Explanations, and Responses**

1. We cited recent relevant works in the related works section.
2. We provided additional details on training and execution in our responses, aiming to clarify the methodology.
3. We have expanded on the descriptions of the visualizations (i.e., Figs. (2) and (3)) in the ablation study section to improve clarity.
4. We provided a more detailed explanation of the concept of 'cooperative-temporal knowledge', how it is learned, and its role in improving multi-agent cooperation.
5. We explained the generalization capability of task-specific skills to unseen tasks.

We hope that these revisions and responses significantly enhance the clarity and quality of the paper and meet your expectations. We would greatly appreciate it if you could consider further raising your ratings of this paper to give us more chances to present at the ICLR conference.

Thank you once again for your dedicated work as reviewers and your support of our paper.

Best regards,

Authors

---

### Public Comment · ~Qifan_Liang1 · 2026-05-23
**question regarding Eq. (9)**

I have a question regarding Eq. (9).

From the inequality in Eq. (9), the left-hand side appears to be an upper bound of the negative KL term rather than a lower bound. However, maximizing this upper bound does not necessarily imply maximizing the right-hand side (or equivalently minimizing the KL divergence).

Does this mean that Eq. (9) cannot rigorously justify the optimization objective used later in Eq. (10)/(11)? In other words, is there a gap between the theoretical derivation and the actual optimization objective?

---

### Meta-Review · Area_Chair_AZ29 · 2024-12-21

**Metareview:**

This paper proposes an approach for learning generalizable multi-agent policies from offline multi-task data. The method jointly learns common skills that capture cooperative behaviors across tasks and task-specific skills for fine-grained action execution. The reviewers generally agree that the paper addresses a challenging and meaningful problem in multi-agent reinforcement learning. They appreciate the novelty of jointly learning common and task-specific skills, the comprehensive experiments showing consistent improvements over state-of-the-art methods. However, some concerns and weaknesses are raised. The main weaknesses identified by the reviewers include the complexity of the algorithm framework compared to previous methods, the need for improved writing clarity, and the lack of clear explanations on how common skills integrate cooperative temporal knowledge and enable offline exploration. There are also questions about the distinctiveness of common and task-specific skills, the representation of multi-agent skills at both temporal and subgroup levels, and the reproducibility of results using the provided code.

**Additional Comments On Reviewer Discussion:**

Initially, reviewers raised concerns about framework complexity, methodological clarity, and experimental validation. The authors provided responses including new experimental results on sensitivity analysis, additional ablation studies on the Stalker-Zealot task set, and comparison with ODIS on MAMuJoCo. This led to constructive dialogue, with several reviewers explicitly increasing their scores based on the authors' thorough responses. The authors' commitment to improving documentation and releasing additional code further strengthened the positive reception.

---

### Decision · Program_Chairs · 2025-01-22

Accept (Poster)